# Overview of Anti-Inflammatory and Anti-Nociceptive Effects of Polyphenols to Halt Osteoarthritis: From Preclinical Studies to New Clinical Insights

**DOI:** 10.3390/ijms232415861

**Published:** 2022-12-13

**Authors:** Laura Gambari, Antonella Cellamare, Francesco Grassi, Brunella Grigolo, Alessandro Panciera, Alberto Ruffilli, Cesare Faldini, Giovanna Desando

**Affiliations:** 1Laboratorio RAMSES, IRCCS Istituto Ortopedico Rizzoli, via di Barbiano 1/10, 40136 Bologna, Italy; 21st Orthopedic and Traumatology Clinic, IRCCS Istituto Ortopedico Rizzoli, via G.C. Pupilli 1, 40136 Bologna, Italy

**Keywords:** osteoarthritis, natural compounds, nutraceuticals, bioactive dietary molecules, polyphenols, inflammation, pain

## Abstract

Knee osteoarthritis (OA) is one of the most multifactorial joint disorders in adults. It is characterized by degenerative and inflammatory processes that are responsible for joint destruction, pain and stiffness. Despite therapeutic advances, the search for alternative strategies to target inflammation and pain is still very challenging. In this regard, there is a growing body of evidence for the role of several bioactive dietary molecules (BDMs) in targeting inflammation and pain, with promising clinical results. BDMs may be valuable non-pharmaceutical solutions to treat and prevent the evolution of early OA to more severe phenotypes, overcoming the side effects of anti-inflammatory drugs. Among BDMs, polyphenols (PPs) are widely studied due to their abundance in several plants, together with their benefits in halting inflammation and pain. Despite their biological relevance, there are still many questionable aspects (biosafety, bioavailability, etc.) that hinder their clinical application. This review highlights the mechanisms of action and biological targets modulated by PPs, summarizes the data on their anti-inflammatory and anti-nociceptive effects in different preclinical in vitro and in vivo models of OA and underlines the gaps in the knowledge. Furthermore, this work reports the preliminary promising results of clinical studies on OA patients treated with PPs and discusses new perspectives to accelerate the translation of PPs treatment into the clinics.

## 1. Introduction

Knee osteoarthritis (OA) is a complex clinical disorder that affects multiple joint tissues, including cartilage, subchondral bone, meniscus and synovium, with a tremendous impact on patients’ quality of life [1]. Metabolic alterations associated with obesity and diabetes, along with ageing and injuries, are among the main risk factors for OA. This heterogeneous framework reflects the variety of phenotypes in OA patients, which contributes to limiting the success of clinical trials on emerging drugs [2]. Joint inflammation is one of the main hallmarks of OA and, together with other pathological processes, ultimately leads to a disruption of joint homeostasis and biomechanics and causes joint pain and stiffness. The main goal of the treatments for knee OA is to act at an early stage in order to halt these degenerative and inflammatory aspects and improve mobility and function. Most of the current strategies include non-steroidal anti-inflammatory drugs (NSAIDs), cyclooxygenase-2 (COX-2) inhibitors and intra-articular (IA) treatments with anti-inflammatory agents, viscosupplements and biologics agents, which provide only partial and temporary pain relief and entail several limitations and risks [3]. Therefore, there is an urgent need to find new, safer alternatives to counteract joint inflammation, degeneration and pain. Recently, bioactive dietary molecules (BDMs) have emerged as potential preventive and therapeutic options to complement conventional treatments in OA [4,5]. Indeed, several studies have emphasized their pleiotropic effects in modulating specific pathogenetic pathways implicated in joint dysfunction. In particular, BDMs belonging to the polyphenols (PPs) family have been described to modulate inflammatory [6] and pain-associated biological targets, making them good candidates to treat symptomatic OA [7,8]. Most of the evidence of their protective role comes from preclinical studies, while the number of clinical studies remains limited [9,10].

This review aims to provide an overview of past and current experimental research focusing on the anti-inflammatory and anti-nociceptive effects of PPs in OA by highlighting their molecular mechanisms of action and the data on their bioavailability, efficacy and safety (when available). Critical gaps in the knowledge and safety of PPs are discussed in order to guide future research and clinical efforts. Finally, this work examines the challenges and future perspectives to improve their applications and biological performance to treat OA.

### Literature Search Strategy

PubMed and Web of Science literature searches were performed with the following keywords: “knee osteoarthritis,” “inflammation,” “pain,” “natural compounds,” “nutraceuticals” and “polyphenols”. Despite their biological relevance, articles focusing on extracts were excluded.

## 2. OA Description: Focus on Inflammation and Pain

### 2.1. Molecular Signalling Pathways during Inflammation in Knee OA

Despite the long-standing definition of OA as a wear disorder, the number of studies focusing on the link between inflammation and OA has been rapidly growing [11]. In this context, the interplay between cartilage and synovium plays a key role in fueling inflammatory responses by boosting the release of inflammatory cytokines, which are directly implicated in joint damage. Another key player and driver of inflammation in OA is the impairment of the immune system [12], particularly the imbalance of M1/M2 macrophages. The up-regulation of the M1 subset has harmful effects due to the synthesis of pro-inflammatory mediators and the release of tissue debris into the synovial cavity [13]. Moreover, this phlogistic circuit has important consequences on the subchondral bone via the up-regulation of osteoclastogenesis, bone resorption and angiogenesis [14]. The inflammation evokes catabolic responses, which result in the decrease in specific markers of cartilage homeostasis, i.e., collagen II, aggrecan, glycosaminoglycans (GAGs), SRY-box transcription factor 9 (Sox-9), β-integrins; and up-regulation of fibrotic and hypertrophic markers, i.e., collagen I, collagen X, runt-related transcription factor 2 (Runx-2), vascular endothelial growth factor (VEGF)-A, matrix metalloproteinase (MMP)-13.

In OA, one of the first pathological changes is the metabolic activation of chondrocytes and their phenotypic shift towards a degradative and hypertrophic-like state. These alterations determine the synthesis of inflammatory mediators and cartilage-degrading enzymes involved in the release of extracellular matrix (ECM) fragments from cartilage, known as damage-associated molecular patterns (DAMPs) [15,16]. Forms of DAMPs present in the OA joint include: (i) alarmins, i.e., high-mobility group box-1 (HMGB1) and S100 proteins; (ii) crystals, i.e., calcium pyrophosphate dihydrate (CPPD) and basic calcium phosphate (BCP). DAMPs interact with different pattern recognition receptors (PRRs): (i) toll-like receptors (TLRs) such as TLR2, and TLR4; (ii) NOD-like receptors (NLRs); and (iii) the receptor for advanced glycosylation end products (RAGEs) [17]. Their binding induces the synthesis of: (i) inflammatory cytokines such as interleukin (IL)-1β (IL-1β), tumor necrosis factor-α (TNF-α), IL-6/-8; and (ii) catabolic mediators (proteases) such as MMP-1,-3,-13, and a disintegrin and metalloprotease with thrombospondin motifs (ADAMTs)-4,-5. Among TLRs, TLR4 is overexpressed in chondrocytes from OA patients more than in non-OA patients [18]. Their binding with ligands induces the translocation of NF-κB into the nucleus, the activation of the MyD88 and the TIR domain-containing adaptor-inducing interferon-β (TRIF) pathways, resulting in the induction of inflammatory responses [19]. The imbalance between MMPs and their negative regulators (i.e., TIMPs, CITED2) is also implicated in cartilage breakdown in OA patients.

Other inflammatory mediators associated with OA include: nitric oxide (NO), catalyzed by inducible NO synthase (iNOS); cyclooxygenase-2 (COX-2), encoded by prostaglandin-endoperoxide synthase 2 (PTGS2); prostaglandin E2 (PGE2); and chemokines such as monocyte chemoattractant protein-1 (MCP-1, also called CCL2), C-C Motif Chemokine Ligand (CCL5); and C-X-C Motif Chemokine Ligand (CXCL)-1 [20].

In addition to inflammation, the production of reactive oxygen species (ROS) is closely associated with the evolution of the severity of OA. NADPH oxidase (NOX) is up-regulated during OA in chondrocytes, which is the major producer of ROS, especially hydrogen peroxide (H_2_O_2_) [21]. H_2_O_2_ induces mitochondrial damage, lipid peroxidation, and DNA damage, leading to the inhibition of ECM synthesis, ECM degradation, chondrocyte apoptosis, and inflammatory cytokines overproduction, which further leads to MMP formation. In turn, lipid peroxidation (of which MDA is a common indicator) activates the nucleotide-binding oligomerization domain-like receptor protein 3 (NLRP3) inflammasome by amplifying the inflammatory circuit via the release of IL-1β and IL-18. Notably, in OA synoviocytes, MDA is increased, while GPX, an anti-oxidant agent, is decreased [22]. Poly (ADP-ribose) polymerase (PARP)-1 is a nuclear enzyme activated during apoptosis (downstream to caspase-3 activation and DNA strand breaks), which plays a crucial role in oxidative stress-induced inflammation. It regulates the production of several inflammatory molecules, including transcription factors, cytokines, chemokines, COX-2, and iNOS [23]. Corroborating the link between ROS and inflammation in OA is the increasing evidence that NRF2, widely renowned as the master gene of anti-oxidant response, plays a pivotal role in the protection of joint cartilage during OA pathogenesis by modulating inflammation [24]. Nrf2 was found to be decreased after IL-1β stimulation [25] and in OA articular cartilage, along with glutathione s-transferase alpha (GSTA)-4, predisposing to cartilage degradation and synovium inflammation increasing hydroxynonenal (HNE) production [26]. Moreover, heme oxygenase (HO)-1, a downstream factor of Nrf2, is a critical factor in Nrf2-mediated NF-κB inhibition [27].

Several signaling pathways are implicated in inflammation during OA, in particular: (i) NF-κB; (ii) phosphoinositide 3-kinase/protein kinase B (PI3K/AKT); and (iii) mitogen-activated protein kinase (MAPK). The NF-κB pathway is widely recognized as the most important signaling regulating inflammation. It can be activated by several cytokines (TNF-α and IL1-β) and other signaling pathways (e.g., MAPK, Nrf2 signaling). The NF-κB protein normally forms a complex with a nuclear factor of kappa light polypeptide gene enhancer in B-cells inhibitor alpha protein (IκBα), which keeps it in an inactive state in the cytoplasm. During inflammation, the IκB kinase (IKK) complex phosphorylates IκB proteins (p65 NFkB, IκBα) and IκBα undergoes proteasomal degradation. The resulting activation of the NF-κB complex leads to the transcription of the genes involved in inflammation (i.e., immunomodulatory molecules, cytokines, COX-2, MMPs, and iNOS) and the imbalance between anabolism and catabolism [28,29]. In addition to its effects on inflammation, NF-κB can also modulate the expression of proteins implicated in apoptosis, such as c-caspase3, Cyto-c, Bax and Bcl-2, with contrasting literature about its role [30].

PI3K/Akt/mTOR signaling can promote both the anabolic and catabolic pathways via the activation of distinct downstream effectors, among which AKT and mammalian target the rapamycin complex 1 (mTORC1) [31]. The MAPK are intracellular Ser/Thr kinases classified into four sub-families: (i) extracellular signal-regulated kinase (ERK 1 and 2); (ii) c-Jun NH2-terminal kinase (JNK 1,2, and 3); (iii) p38 (α, β, γ, δ); and (iv) ERK5. MAPK has a crucial role in modulating multiple pathways implicated in joint destruction.

Several epigenetic factors (induced by various environmental factors i.e., pollutants, diet, exercise, stress) have been identified as playing a key role in OA inflammation and pain. They include DNA methylation, histone modifications and non-coding RNAs (i.e., small interfering RNA, miRNA, and long non-coding RNA) [21,32]. A growing body of research reported altered DNA methylation associated with specific genes involved in OA, such as MMP-3, -9, -13, and ADATMS-4 [33]. In this line, the aberrant expression of miRNAs, in terms of up-regulation (i.e.,miR146, miR34a, miR155, etc.) or down-regulation (i.e., miR124, miR-130, miR-140, miR-210, etc.), was documented in OA [34]. Similarly, histone-modifying proteins display important roles in the pathogenesis and progression of OA. In particular, sirtuins (SIRT-1,2,3,4,5,6,7) belonging to class III of histone deacetylases (HDACs) have a great impact in regulating chondrocyte differentiation and functions [35]. The silent information regulator of transcription 1 (SIRT1) is a longevity gene involved in the deacetylation of histones and transcription factors with pleiotropic activity [36]. Extremely relevant in the context of OA, SIRT1 is a negative regulator of inflammation, and in particular, inhibiting the P38-mediated inflammatory signaling pathway. The TLR4/MyD88/NF-κB signaling pathway reduces multiple proinflammatory cytokines and chemokines. Notably, the expression of the SIRT1 is poorly expressed in OA patients [37] and it is cleaved and inactivated following the treatment of chondrocytes with inflammatory stimuli [38,39,40].

Regarding the ageing and metabolic phenotypes of OA, patients display persistent low-grade systemic inflammation. During inflammaging, OA patients display increased levels of advanced glycation end-products (AGEs) and phlogistic mediators, which generate a senescence-associated secretory phenotype (SASP) [14]. Notably, ageing reduces the expression of HO-1, thus generating a reduction in anti-oxidant defenses and a pro-inflammatory tendency which generates ROS and MMPs, causing cartilage ECM degradation and joint dysfunction [41].

During obesity, metabolic dysfunction is the main pathophysiologic factor linked to inflammation (meta-inflammation). In this regard, several adipose-tissue-derived cytokines (adipokines) are secreted, among which leptin, which activates the JAK-STAT3 signaling pathway, along with increased expression of the interleukin-1 receptor-associated kinase (IRAK)-1 and IRAK4, activate tumor necrosis factor receptor-associated factor 6 (TRAF6) and NF-κB [28].

Recently, a body of research gave evidence of the presence of a gut-joint axis in the OA pathogenesis due to a disruption of gut homeostasis with the display of an inflammatory phenotype (dysbiosis), causing an alteration to the microbiota composition. The translocation of bacteria toward the joint would seem to be implicated in dysbiosis and contribute to the pathogenesis of OA, even if further investigations are still necessary [42].

Overall, the molecular understanding of factors driving OA is key to identifying alternative candidates for targeting inflammatory pathways as potential novel disease-modifying therapies to treat OA.

### 2.2. Biological Basis of Pain in Knee OA

Severe joint pain is one of the main symptoms of OA, limiting simple daily activities and negatively affecting patients’ mental well-being with a consequent impact on their quality of life. In the clinic, the most commonly used validated criteria to assess pre- and post-treatment of pain in OA include the visual analogue scale (VAS), the Western Ontario and McMaster Universities Arthritis Index (WOMAC) pain subscale, and the Knee injury and Osteoarthritis Outcome Score (KOOS) [43]. These scores are aimed to track how OA patients experience pain in their daily (i.e., walking, using stairs, in bed, etc.), sports and recreational activities. Despite their widespread use, none of them can adequately describe the complex pain experienced by OA patients nor guide analgesic therapies [43]. The main structural changes associated with pain in OA, identified by MRI studies, include bone marrow lesions, synovitis, knee effusion, periarticular lesions, meniscal tears, and bone lesions [44].

Depending on the type of stimuli, pain is classified into three types: nociceptive, neuropathic and inflammatory. The aetiology of OA-related pain is still poorly understood for the relative paucity of mechanistic studies, even if the pathogenesis of pain is primarily thought to be inflammatory.

Nociceptive pain is a typical response of the nervous system to noxious stimuli, particularly following physical/mechanical injuries of tissues or chemical stimuli (i.e., heat, chemicals, inflammation, pressure). This is an acute form of pain, which resolves after tissue healing or the removal of noxious stimuli. Joint nociceptors (pain-sensing afferent neurons) are primarily located in the bone [45], synovial membrane [46], menisci [47] and soft tissues [48], while they are not present in the cartilage, which lacks nerve innervation unless ectopically innervated [49]. In response to noxious stimuli, nociceptor activation leads to the transduction of pain signals through interactions between the immune and nervous systems and the release of inflammatory mediators and neuropeptides [50]. There are several nociceptor-specific mediators involved in joint pain [51]. Among these, transient receptor potential (TRP) channels (ion channel receptors) mediate pain sensations by stimulating the neural secretion of calcitonin gene-related peptide (CGRP) and substance P (SP) secretion [52].

Neuropathic pain, often associated with allodynia (pain in response to normally non-noxious stimuli), is produced through damage to the central or peripheral nervous system. This is a chronic form of pain usually uncontrolled by analgesics. In peripheral neuropathic pain, the biological mechanisms underlying a nerve injury include the increased expression of voltage-gated sodium channels and new connections between adjacent neurons (“sprouting” or “ephaptic crosstalk”). These processes cause an increased release of: (i) neurotransmitters from the dorsal horn (i.e., glutamate, SP, gabaminergic, adenosine, 5 HT1, 5HT3, etc.); and (ii) immune and inflammatory mediators (i.e., catecholamine, histamine, cytokines, prostaglandins and ATP), which are responsible for nociceptor sensitization. In central neuropathic pain, the abnormal stimulation of neurons leads to an increase in N-methyl-D-aspartate receptor (NMDA) receptors and the up-regulation of cyclooxygenase and purinergic P2X3 receptors. Damage to the peripheral and central nervous system causes an abnormal generation and transmission of impulses. Inflammatory and metabolic disorders are among the major causes involved in neuropathic pain, which occur as a result of damaging stimuli [53].

Inflammatory pain is a response to chronic local inflammation in the joint and it is strongly interconnected to both nociceptive and neuropathic pain. Crosstalk and bidirectional interactions between the immune system and nociceptive neurons are central to inflammatory pain. Painful inflammatory stimuli can cause allodynia, hyperalgesia (neuronal hypersensitivity), mechanical sensitization and structural neuroplasticity of joint nociceptors.

Pain-related inflammatory mediators include IL-6 and TNF-α, which increase the serum levels of C-reactive protein (CRP), a circulating marker of systemic inflammation that correlates with increased pain in OA. Recently, Nees TA et al. provided a broad profile of inflammatory mediators of potential clinical relevance associated with joint pain in the synovial fluid, finding that IL-10, IL-12, IL-13, SCGF-β, and VEGF show clear correlations with joint pain and function [50]. Similarly, Liu et. Al. have indicated that blocking the IL-17 signaling pathway might contribute to treating OA pain due to the strong correlation between IL-17 levels and OA pain based on the WOMAC scale [54]. Increased levels of inflammatory mediators promote the expression of proteolytic enzymes, such as MMPs and ADAMTs [55].

Two key pain-sensitizing molecules implicated in OA pain are nerve growth factors (NGF) and CCL-2 [56]. NGF is produced by macrophages, mast cells, synoviocytes, and neutrophils and is up-regulated by cytokines such as TNF-α. The binding of NGF with the receptor tropomyosin receptor kinase (Trk)A leads to the up-regulation of the expression of the ion channels transient receptor potential vanilloid (TRPV)-1 and Nav1.8, CGRP, SP and brain-derived neurotrophic factor (BDNF) [57].

Among the main neurotransmitters implicated in joint pain, it is possible to distinguish between both: (i) inflammatory mediators, such as PGE2, PGI2, LTB4, NGF, proton, BK, ATP, adenosine, SP, neurokinin (NK)-A, B, 5-HT, histamine, glutamate, nor-epinephrine (NE) and NO; and (ii) non-inflammatory mediators, such as CGRP, gamma-aminobutyric acid (GABA), opioid peptides, glycine and cannabinoids. They determine the production of a second messenger that interacts with several ion channels by regulating pain [53].

MAPK and PI 3-K/AKT/mTOR, wingless-related integration site (Wnt) signaling pathways, are thought to be among the primary pathways implicated in chronic pain [31,58]. Under inflammatory and neuropathic pain, MAPK pathways are activated in neuron cells and regulate pain sensitization through central and peripheral mechanisms. Indeed, inhibitors of ERK, p-38 and JNK are effective in relieving pain symptoms, but cannot be used as drugs due to several issues [59]. the up-regulated expression of mTOR correlates with the decreased synovitis, strongly associated with pain [60]. The activation of canonical Wnt signaling has multiple effects on different tissues in the OA joint: (i) osteophyte formation; (ii) cartilage destruction; and (iii) synovitis, with consequent pain symptoms in OA patients [61]. However, how Wnt/β-catenin signaling regulates inflammatory cytokines and chemokines in the DRG and dorsal horn of the spinal cord, and the brain, and the crosstalk of Wnt/β-catenin signaling with the immune system is yet to be elucidated.

Other pathways closely associated with OA pain are the hypothalamic-mediated neuromodulation pathway (related to leptin, the neuropeptide Y (NPY) system and other neuropeptides [62] and the endocannabinoid (EC)-related pathway (which involves the frontal cortex, nucleus accumbens, striatum and hippocampus and is associated with chronic OA pain) [63].

Most of the progress in understanding the biological mechanisms of pain and its associated genes and molecules (neurotransmitters, receptors, intracellular messengers) comes from preclinical animal studies. To this end, rodent models have extensively contributed to the comprehension of several aspects related to acute and chronic pain states [64]. In particular, the mono-iodoacetate (MIA) model is often used to induce rapid pain-like responses as it induces the alteration of chondrocyte glycolysis and, consequently, cell death, vascularization processes, bone necrosis, bone collapse, and inflammation. The release of inflammatory mediators plays a crucial role as they fuel the pain response through their interaction with nociceptors upon tissue injury [65]. Other animal models of inflammatory pain include: (i) injection of capsaicin into the joint; (ii) carrageenan injection; (iii) complete Freund’s adjuvant (CFA) into the tail, paw and joint. Reflexive and non-reflexive pain tests are the two main types of outcome measures adopted in preclinical studies to assess pain [66,67,68]. Reflexive pain tests include the application of noxious stimuli (thermal/mechanical) at the site or outside the site of injury, with the consequent activation of nociceptors. In this regard, the assessment of paw withdrawal latency (PWL), paw withdrawal threshold (PWT), mechanical withdrawal threshold (MWT) and thermal withdrawal latency (TWL) are among the main indices considered in preclinical studies [66,67]. Non-reflexive pain tests include measures of spontaneous pain behavior, avoidance of evoked stimuli and quality assessment of life and function [68].

In addition, the establishment of the most appropriate animal model, which closely mimics the human experience of pain, would be a powerful tool to gain a full understanding of the potential use of future therapies to guide clinicians in the treatment of OA [69].

## 3. Traditional Therapeutic Strategies to Counteract Inflammation and Pain in OA: Pros and Cons

Although there have been successes in preclinical and early clinical studies, phase three clinical trials have failed so far and there are still no approved disease-modifying treatments (DMOAD) on the market.

The therapeutic strategies adopted for the treatment of OA are closely related to the staging of the disease. In the absence of DMOAD (targeting the underlying disease pathogenesis), the gold standard therapy for a mild-moderate grade of OA is still limited to the temporary systemic relief of symptoms through the use of analgesics, NSAIDs (meloxicam, diclofenac, naproxen), corticosteroids, paracetamol, COX-2 inhibitors (celecoxib); combined or not with the topical application of creams [70,71]. Traditional NSAIDs and COX-2-selective NSAIDs (coxibs) promote analgesic effects via the inhibition of the COX family of enzymes, implicated in the synthesis of prostaglandins, potent inflammatory and hyperalgesic mediators [72]. NSAIDs show beneficial effects through the inhibition of inflammation and pain; however, these oral medications often have side effects, including gastrointestinal disorders (e.g., irritation of the gastrointestinal tract, peptic ulcers, intestinal bleeding), immune reactions, toxicity, and cardiovascular effects [73]. As OA is a chronic condition that requires a long duration of treatment, this limits the success of these intervention approaches. Conversely, topical forms of NSAIDs may offer an alternative solution to relieve OA pain by overcoming the issues correlated to oral formulations; however, they only provide pain relief in patients with mild-to-moderate OA [74] and are recommended as an initial option, particularly for elderly patients [75]. Oral opioid administration is an option with limited application, not only due to its prolonged use often leading to physical dependence, but also because it exerts only minimal relief of OA symptoms within 12 weeks, as demonstrated by a recent meta-analysis of randomized controlled trials on patients with knee and/or hip OA [76,77]. Due to the localized nature of OA, injectable preparations are an attractive treatment approach to: (1) provide mechanical stability and lubricating mechanisms in the synovial fluid; (2) exert a direct effect on inflammation, oxidation and proteases production [78,79]; (3) mediate IA drug injection, minimizing side effects. IA therapies, include viscosupplements (hyaluronic acid, HA), corticosteroids (dexamethasone, methylprednisolone acetate, triamcinolone acetate) and blood-derived products to treat patients with mild to moderate symptomatic knee OA.

There are several formulations of HA (different sources, molecular weights, and purification methods) that are widely employed in clinics, even if they have shown some limitations, such as temporary pain relief and the need for repeated administrations. Its mechanism of action includes: (i) improvement of joint lubrication; (ii) reduction in collagen degradation; (iii) promotion of anabolic molecules; and (iv) reduction in several catabolic and inflammatory mediators [80,81,82]. Corticosteroids exert an immediate reduction in the patient’s pain with short-term effects, however, they have undesirable side effects when administered at high doses and frequency [83,84]. Platelet-rich plasma (PRP) IA injections improved short-term pain and knee function scores in a single-center prospective randomized controlled study with a one year follow-up and were safer than corticosteroids treatment [84]. However, the results of the use of PRP are extremely heterogenous, as its activity depends on the donor, number of platelets, type/quantity of growth factors, and the method of preparation, which makes it impossible to draw firm conclusions.

Despite all the progress in the standard conservative treatments, these approaches are often insufficient in controlling the patient’s symptoms, leading to the surgical indication of a knee joint replacement. This poses the need to search for alternative strategies to halt inflammation and pain with minimal side effects and more homogenous formulations to ensure repeatability in the clinical setting.

Among the new candidates as alternative strategies, dietary interventions are emerging as key non-pharmacological approaches for preventing and treating OA and have recently attracted increasing attention from nutritionists, food scientists, and even consumers, for their recognized roles in human health, particularly by inhibiting inflammatory processes.

## 4. Efficacy and Safety of PPs in the Management of OA

### Role of PPs in Modulating Inflammation and Pain: Focus on Preclinical In Vitro and In Vivo Studies

PPs are natural compounds with phenolic structural features categorized into several classes according to their chemical composition, which influence their stability, bioavailability, and physiological functions [85]. The major known classes of PPs are phenolic acid, flavonoids, lignans, and stilbenes, which are shown in Figure 1, along with their main PPs constituents [85]. PPs are found abundantly in a wide variety of foods (fruits, vegetables, cereals, spices, herbs) and beverages (coffee, tea, wine, chocolate) and their use has been shown to protect against several chronic pathologies. Thanks to their pleiotropic effects, PPs exert several benefits on human health [86]. Moreover, they have recently been defined as epigenetically active dietary components that can be used as therapeutic interventions in alleviating persistent inflammation by targeting the epigenome [87].

Several authors have investigated in vitro the effect of PPs on inflammation in OA [25,26,88,89,90,91,92,93,94,95,96,97,98,99,100,101,102,103,104,105,106,107,108,109,110,111,112,113,114,115,116,117,118,119,120,121,122,123,124,125,126,127,128,129,130,131,132,133,134,135,136,137,138,139,140,141,142,143,144,145,146,147,148,149,150,151,152,153,154,155,156,157,158,159,160,161,162,163,164,165,166,167,168,169,170,171,172,173,174,175,176,177,178,179,180,181,182,183,184,185,186,187,188,189,190,191,192,193,194,195,196,197] (Table 1), particularly on articular chondrocytes obtained from OA joints, whereas a few studies have been carried out on peripheral blood mononuclear cells, osteoblasts, macrophages, and lymphocytes. Most studies have tested the preventive (treatment before OA induction- cell pre-treatment) or therapeutic (treatment after OA induction) effects of PPs on cell models mimicking OA inflammation. Among them, the most employed is the stimulation with IL-1β, due to its pivotal role in the pathogenesis of OA. In chondrocytes, IL-1β up-regulates the expression of several inflammatory mediators (e.g., NO, PGE2, IL-6) and inhibits ECM synthesis (collagen-II and aggrecan) by promoting the production of proteases (e.g., MMP-3, MMP-13, ADAMTS-4/5). Further in vitro models of inflammation include the use of other stimuli, such as AGE, TNF-α, H_2_O_2_, tert-butyl hydroperoxide (TBHP, a more stable form of H_2_O_2_ which activates ROS and ER), lipopolysaccharide (LPS, lipid A and polysaccharides which induce IL-1β and TNF-α expression). Notably, a few studies have tested PPs on co-culture systems between chondrocytes and synoviocytes or monocytes in order to study the crosstalk among joint cells. In general, all of the PP compounds displayed various anti-inflammatory, anti-oxidant and anti-catabolic effects, particularly via the modulation of NF-κB and MAPK signaling pathways, as shown in Table 1.

The most studied PP is resveratrol, belonging to the class of stilbenes found in the red variety of grapes (and consequently red wine), peanuts, blueberries, pines, rhubarb, and root extracts of the weed *Polygonum cuspidatum*. In chondrocytes, resveratrol exerts anti-inflammatory effects via the reduction in several inflammatory cytokines (IL-1β, -18, -6, TNF-α) and proteolytic enzymes (MMP-1, -3, -9, -13, ADAMTS4, ADAMTS5) and through the promotion of typical ECM proteins (collagen II, aggrecan, Sox-9, β1-integrin, GAG). Other molecules modulated by resveratrol are CXCL1, NOX4, PTGS2, and iNOS (Table 1). In the majority of studies, resveratrol administration has been linked to the inhibition of IkBα degradation, NF-kB activation and nuclear translocation. Several authors have demonstrated that resveratrol inhibits TLR4 and, consequently, TLR4/My88/NFkB signaling [88,89,90]. In particular, it targets IKK-NFKB and MAPK/AP-1 signaling pathways [91] and inhibits the production of mature IL-1β [92]. Interestingly, resveratrol modulates the epigenome by activating SIRT1 [93], which drives the inhibition of NFkB via SIRT1/FOXO signaling [94,95]; Yi and al. showed that resveratrol inhibits miR-210-5p, which targets LINC00654 and OGFRL1 [96]. Employing a co-culture model with macrophages and chondrocytes, Limagne et al. demonstrated that resveratrol can inhibit the inflammatory loop through the down-regulation of STAT3 in macrophages and the inhibition of NF kB in chondrocytes [97]. Further evidence of its anti-inflammatory role comes from the Siard et al. group, who demonstrated a decrease in INF-γ and TNF-α in lymphocytes treated with resveratrol [98].

Other PPs largely studied are epigallocatechin-3-gallate, quercetin, hesperitin and icariin, all belonging to the class of flavonoids.

Epigallocatechin-3-gallate is the major catechin found in green tea and exerts an anti-inflammatory effect by targeting: (i) in chondrocytes, the PTEN [99], MAPK [100,101], JNK [102] pathways; and (ii) in synoviocytes, the inflammatory cascades induced by DAMPS CPP crystals [103]. Moreover, it modulates miRNA: it down-regulates miR-29b-3p [99] and up-regulates miRNA-199a-3p [101].

Quercetin is ubiquitously present in fruits and vegetables; it exerts an anti-inflammatory effect through SIRT1/AMPK activation and ER stress inhibition [104], suppressing IRAK1/NLRP3 [105] and p38/MAPK [106] in chondrocytes. Notably, Hu et.al. showed, in a co-culture model of monocytes and chondrocytes, that quercetin: (1) drives the M2 polarization of monocytes; (2) induces anti-inflammatory, anti-catabolic, and anti-apoptotic effects on chondrocytes; (3) promotes the pro-chondrogenic environment via the release of transforming growth factor β (TGF-β) and insulin growth factor in monocytes [107]. The main effector of these processes is the suppression of AKT/NF-κB signaling [107]. Li et. al. showed that quercetin could suppress the expression of the IRAK1/NLRP3 pathway and decrease the levels of pro-inflammatory cytokines (IL-18, and TNF-α) in IL-1β-induced rat chondrocytes [105]. Gain-of-function assays showed that the molecular overexpression of NLRP3 or IRAK1 significantly reversed the anti-inflammatory and anti-apoptotic effects of quercetin in IL-1β-induced rat chondrocytes, whereas NLRP3 knockdown restored its protective function [105].

Hesperetin can be found in some citrus species and exerts an anti-inflammatory response through NF-κB inhibition [108], particularly via the down-regulation of TLR-2 [109]. Interestingly, hesperidin inhibits the expression of IL-17, one of the main cytokines involved in OA pain [110].

Icariin, found in *Epimedii herba*, *Epimedium sagittatum* plants, exerts an inflammatory response through NF-κB inhibition by acting on several pathways: (i) NLRP3 inflammasome-mediated caspase-1 signaling [111]; (ii) TLR4/Myd88 [112,113]; and (iii) p38/ERK/JNK [114]. Liu et al. found that the effects of icariin are associated with the activation of miR-206 in chondrocytes [115]. In LPS-treated synoviocytes, icariin inhibited ferroptosis through the activation of the Xc-/GPX4 and NRF2 axis [116]. Notably, icariin modulates OA pain by: (i) decreasing NPY, NPY1R, SP R and 5-HT1B R neuropeptides; (ii) increasing VIP neuropeptides; and (iii) enhancing the expression of sensory-related genes [117]. To our knowledge, this is one of the few shreds of evidence of PPs modulating key molecular pathways of pain.

As for the other PPs, they act similarly to those above-mentioned in inhibiting inflammation. In particular, several other PPs inhibit NFkB signaling: i.e., carnosol [118], CAPE [119], tangeretin [120], nobiletin [121], naringenin [122], theaflavin-3-3′-digallate [123], cyanidin [124,125], ellagic acid [126]. Similarly to resveratrol and quercetin, other PPs modulate epigenome by acting as SIRT1 inductors: i.e., ferulic acid [127], fisetin [128], procyanidins [129]; while cyanidin was showed to increase SIRT6 [125]. Moreover, other PPs modulate miRNA: carnosic acid (CA) up-regulates miR140 as shown by Ishitobi et. al.; curcumin up-regulates miR-124 and miR-143 [130].

Similarly to quercetin, vanillic acid [131], chlorogenic acid [132], carnosol [118], biochanin [133] and pinoresinol diglucoside [134] induce the expression of TIMP-1. Some PPs such as sesamin [135], and carnosol [118] can suppress the autocrine signaling of IL-1β in chondrocytes.

Moreover, one of the most documented pathways up-regulated by PPs is the NRF2/HO1; i.e., sinapic acid [136,137], xanthohumol [138], myricetin [139], lutein [140], theaflavin-3-3′-digallate [123], delphinidin [141], 6-gingerol [26], CAPE [119], genistein [25] and tangeretin [120]. In particular, it has been reported that Nrf2 and HO-1 could alleviate inflammation through the inhibition of p65 translocation, thus inhibiting NFKB pathways [27,119,120]. Notably, CAPE and lutein directly interact with the Keap1-NRF2 complex by inhibiting ubiquitination, resulting in Nrf2 stabilization and promoting the nuclear translocation of NRF2 [25,140]. Moreover, xanthohumol promotes the binding of HO-1 and C/EBPβ and inhibits C/EBPβ nuclear translocation [142]. The relevance of Nrf2 and/or HO1 signaling, upstream to NF-κB signaling inhibition, was proven by loss of function experiments using silencing [119,120,123,138,142]. Notably, Zhou et al. depicted the interaction among NRF2 and AMPK signaling upon luteolin stimulation [140]. Luteolin exerts anti-inflammatory and anti-catabolic effects on H_2_O_2_ chondrocytes, mediated by Nrf2 and AMPK cascade activation [140]. Similarly, Pan et. al. showed that myricetin activates Nrf2/HO-1 downstream to PI3K/Akt [139].

Other authors have shown the activation of several pathways downstream to PPs activation in chondrocytes. Interestingly, Shi et. Al. found that tangeretin induces an anti-inflammatory effect and prevents OA, due to NF-κB inhibition, by simultaneously modulating Nrf2-HO-1/NF-κB and MAPK/NF-κB signaling pathways [120]. Teng et al. showed that theaflavin-3-3′-gallate inhibited the PI3K/AKT/NF-κB and MAPK pathways while promoting the Nrf2/HO-1 pathway [123]. Chuntakaruk et. al. showed that several anthocyanins can exert anti-inflammatory effects upon AGE stimulation by targeting NF-κB and MAPK signaling [143]. Huang et. al. showed that vanillic acid exerted an anti-inflammatory effect on IL-1β by targeting both MAPK and PI3K/AKT/NF-κB [125]. Shi et. al. showed that tangeretin activates both Nrf2 and MAPK signaling [120]; Chen et al. demonstrated that rosmarinic acid activates both NF-κB and MAPK [144]. Moreover, Kim et al. showed that fisetin activates both the JNK and NF-κB pathways in monocytes [145]. However, the reciprocal interaction among the pathways was not analyzed.

Overall, most of these in vitro studies have provided evidence of the anti-inflammatory effects of the different classes of PPs, particularly after stimulation with IL-1β, as well as their benefits in reducing catabolic factors and up-regulating anabolic factors to avoid cartilage breakdown with minor evidence for pain process [133,135,146].

Biological benefits observed in the in vitro studies were confirmed in different preclinical animal models of OA (Table 2). In vivo models adopted to study the effect of PPs on inflammation and pain included: (i) post-traumatic OA models through surgical approaches (destabilization of the medial meniscus (DMM), anterior cruciate ligament transection (ACLT), etc.) to assess regeneration; and (ii) pain models (MIA-induced OA model, injection of capsaicin or carrageenan injection, complete Freund’s adjuvant) to explore pain response. Most studies have been conducted in mouse and rat models with different ranges of dosages/intervals of PPs and follow-ups to test the safety and efficacy of PPs in counteracting OA. In addition to the monotherapeutic approach, several studies investigated different combinations of PPs with cell-based therapies, or with other classes of BDMs or novel delivery systems (Table 2). Among PPs, resveratrol, quercetin, formononetin, protocatechuic and hesperetin combined with several delivery systems reported chondroprotective (reduced OARSI score, increased collagen II), anti-inflammatory (reduction in synovitis, reduction in inflammatory mediators) and immunomodulatory (polarization towards M2 macrophage subset) effects, thus representing valid tools to reduce pain and inflammatory responses. The oral gavage of punicalagin has been shown to inhibit inflammatory injury in an OA rat model by activating the Foxo1/Prg4/HIF3α axis and the modulation of Foxo1-autophagy-related gene (ULK1, Beclin1, LC3 and p62) [147]. Repeated IA and intraperitoneal injections of quercetin in surgical models of OA showed that it exerts chondroprotective and anti-inflammatory effects in terms of: (i) reduced OARSI score, apoptotic and phlogistic mediators; and (ii) increased anabolic markers (COLL-2, aggrecan) [105,107]. Moreover, Hui et al. [107] validated the immunomodulatory effect of resveratrol in a rat model and described its potential to drive M2 polarization in the synovial membrane. Further evidence of the chondroprotective role of PPs comes from the study of Wei B. et al. group, who reported, in a post-traumatic rabbit model of OA, the inhibition of cartilage degeneration via a decreased ratio of MMP-13/TIMP-1 [148]. To overcome the limitations related to the low bioavailability of quercetin in the articular joint, some authors explored the IA injection of quercetin encapsulated in an intelligent smart gel (thermogel). They described its prolonged activity in situ, associated with the relief of pain symptoms and the delay of OA progression [149]. Similarly, Britti D et al. employed a novel nanocomposite formulation of palmitoylethanolamide and quercetin to ensure a long-lasting effect in carrageenan paw oedema and MIA-induced OA models in rats. In particular, the authors demonstrated that quercetin counteracts inflammatory and pain responses by reducing: (i) inflammatory and catabolic mediators; (ii) paw oedema, thermal hyperalgesia, and mechanical allodynia; and (iii) neurotrophins, such as NGF [150]. In addition to IA and intragastric deliveries, the topic administration of quercetin combined with nanoparticles also reported benefits in inhibiting OA progression via the decrease in proteases such as MMP-9, -13 and ADAMTs-5 [149].

Similar to quercetin, an IA injection of resveratrol into post-traumatic and pain models of OA reported chondroprotective effects [95]. In both models, resveratrol increased the levels of SIRT-1, severely decreased during OA progression, and was involved in the shift towards a hypertrophic phenotype of chondrocytes [95]. In particular, Deng Z et al. demonstrated that the overexpression of SIRT-1 leads to the repression of i-NOS and MMP-13 in the articular cartilage, in part through the modulation of the NF-kB signaling pathway [95]. Regardless of the route of administration (IA, intragastric, oral supplementation), resveratrol displayed anti-inflammatory and anti-nociceptive effects in the MIA-induced OA models [95].

Several studies with different curcumin formulations have demonstrated its role in slowing down OA progression. The oral administration of curcumin counteracted OA progression, but could not affect OA-related symptoms in a DMM-induced OA model. Notably, the topic administration of curcumin encapsulated in customized nanoparticles provided the first evidence of its effect in ameliorating OA-related pain through the reduction in: (i) adipokines and inflammatory mediators; (ii) synovitis and; (iii) tactile hypersensitivity [151]. Taken together, these results suggest that orally delivered curcumin cannot reach biologically/pharmacologically active concentrations in the serum, synovial fluid, or joint tissues to attenuate OA-related pain. In contrast, the combination of curcumin with nanoparticles provided greater benefits in slowing down OA than oral administration. Along this line, Yabas M et al. proposed a highly bioavailable formulation of curcumin, known as Next Generation Ultrasol Curcumin (NGUC) and explored its impact on pain response in the MIA-induced OA model [152]. The authors demonstrated that NGUC promoted anti-nociceptive effects and reduced the levels of antioxidant enzymes SOD, CAT, and GPX, improving the pathophysiology of OA [152]. Recently, Qiu et al. showed the significant benefits, in a mouse model, of OA after treatment with exosomes derived from curcumin-treated MSCs. In particular, this treatment led to the up-regulation of miR-124 and miR-143, which modulate the ROCK1/TLR9 and NF-kB signaling pathway [130].

Both malvidin and naringenin using MIA-induced OA models described anti-nociceptive and pain-relieving effects [122,153]. Notably, naringenin treatment showed pain alleviation from the fourth day of the treatment and showed a dose-dependent effect (the pain alleviation was more pronounced at a dose of 40 mg/kg rather than 20 mg/kg), as assessed by PWL and PWT in MIA-induced OA rats [122]. In general, all the evaluated PPs in post-traumatic models of OA displayed a significant reduction in inflammatory and catabolic genes and an up-regulation of anabolic markers. The treatment of PPs in MIA-induced OA models, carrageenan paw oedema, and obesity-related models allowed OA pain alleviation and a reduction in inflammatory mediators, implicated in the onset of neuropathic and inflammatory pain (Table 2). Indeed, combing PPs with new technological approaches showed a high bioavailability at the site of injury and an improvement in the biological response to halt OA-associated processes.

Overall, these animal studies have produced further indications of the protective role of PPs in halting OA progression through various mechanisms of action. Furthermore, they provided more evidence of their anti-nociceptive properties through the use of reflex tests combined with non-stimulus evoked methods, such as weight bearing and gait analysis. However, the different conditions used in preclinical in vivo studies, including various severities of OA, different models, routes of administration, doses, dosing intervals and follow-up, offer a multifaceted scenario in which clear recommendations to better guide clinical choices have not yet been defined (Table 2 and Figure 2).

## 5. From Basic Research to Translational Applications of PPs

### 5.1. Clinical Studies on PPs for the Management of OA

Recently, interest in natural compounds has grown tremendously worldwide, leading to an increase in the consumption of these products as an alternative to treating OA in the early stages of the disease. The BDMs market has recorded staggering sales rates due to increased patient awareness of the benefits of taking natural compounds with minimal side effects compared to conventional drugs, such as NSAIDs [217,218]. The aforementioned preclinical in vitro and in vivo studies have shown that PPs may represent adjuvant strategies to treat OA by acting on several signaling pathways involved in the release of inflammatory and catabolic mediators (Table 1 and Table 2). However, only a few studies have been performed, thus hindering their widespread clinical application (Table 3). Among the PPs class, curcumin, resveratrol, quercetin, Pycnogenol^®^ and capsaicin have been included in several clinical studies for the treatment of OA. Most of these studies have shown that their supplementation by different routes of administration (IA, topic, oral) can improve pain symptoms by reducing inflammatory mediators. In particular, OA patients treated with resveratrol monotherapy (oral dose of 500 mg per day) reported a reduction in the VAS and KOOS scores and good safety and tolerability, based on liver and renal function tests and haematological indices [219]. Similarly, studies on pycnogenol^®^ and quercetin showed benefits in pain relief (VAS score) thanks to their ability to reduce all the major inflammatory actors involved in OA. The feasibility of combining PPs with other types of molecules, such as other families of BDMs or classic anti-inflammatory drugs, is growing with the aim of improving their efficacy and reducing drug-related side effects. It has been proven that PPs can work as adjuvants by reducing the number of doses of drugs, with enormous benefits for OA patients. In this regard, the synergistic effect of resveratrol, added to meloxicam therapy for knee OA, has been shown to reduce pain more effectively than meloxicam alone [220]. Moreover, a significant reduction in serum biomarkers of inflammation implicated in pain response, such as TNF-α, IL-1β, IL-6, and complement proteins has been noticed [220] (Table 3). Despite the promising results on the use of PPs, the heterogeneity of the existing literature and the scarcity of randomized, controlled clinical studies in humans make it difficult to propose specific recommendations on the amount of PPs to use as nutritional recommendations in personalized therapeutic approaches.

### 5.2. PPs: Research and Clinical Gaps in OA

Considering both the upsides and downsides, the future use of BDMs seems promising, as they would ideally ensure a high safety profile and significant therapeutic benefits to treat OA patients. In this context, the strong interaction between the knowledge derived from basic research and clinical needs is key to promoting solutions capable of ensuring the well-being of patients. Basic research is crucial to substantiate the biological efficacy of BDMs, providing evidence-based medicine. On the other hand, the clinicians’ perspective is fundamental to focus on the patients’ needs and identify potential limitations of BDMs. To bridge the gap between research and the clinic, and to overcome the current shortcomings that limit their effectiveness, several key aspects need to be refined.

First, the development of products with safe and effective roles is necessary to ensure successful clinical outcomes. In particular, the absence of local and systemic toxicity (cytotoxicity, mutagenicity and genotoxicity), and non-immunogenic (structural features, contaminants, dose and length of treatment) reactions are among the main gaps in this field of research. Despite much evidence on the benefits of PPs in OA, the number of studies focused on their safety profile is low and sometimes with little statistical evidence due to the unavailability of large-scale evaluations in clinical trials [227,228]. Apart from the beneficial effects of PPs, there are controversial data on their role at high dosages, which require more detailed efficacy studies [229]. Pesticides represent one of the most harmful contaminants in BDMs because of their severe toxicity following their ingestion (skin rash, respiratory neurological disorders).

Second, the lack of standardization of products is another major roadblock, which may alter their biological properties and introduce certain risk factors that could undermine their stability. Differences in the region of origin, harvest period and cultivation of PPs may be elements involved in the heterogeneity of products and, consequently, in their altered efficacy. Moreover, the presence of different formulations (tablets, beverages, etc.), and batch-to-batch variations have not allowed us to obtain clear indications of their therapeutic value for the treatment of this disorder.

Third, in terms of their exert biological effects, BDMs need to reach the affected joints; this highlights their main limitation: oral bioaccessibility and bioavailability. Bioaccessibility (fraction of a compound released from food by dietary supplementation to ensure absorption) and bioavailability (fraction of a compound that reaches its site of action) are key prerequisites to ensure the presence of BDMs at the site of action. Ideally, these two properties should be maximized to obtain their best efficacy. Bioaccessibility and bioavailability depend on several factors: route of administration, chemical stability, microenvironment conditions, matrix interactions, and gut microbiota. Regarding the route of administration, topical delivery has advantages over conventional routes (dietary supplementation); in particular, it avoids first-pass metabolism and is a non-invasive mode of drug delivery with a sustained and controlled release profile. Furthermore, the targeting of cartilage via oral administration has been questioned due to the systemic application of a therapeutic agent to the relatively avascular nature of articular cartilage [227]. In this regard, capsaicin, a polyphenol compound, is not suitable for oral administration due to its high first-pass metabolism and gastric irritation. Aqueous aloe vera gel and Carbopol 934 with capsaicin in clove oil emulsion improved the permeability properties [228,229]. To date, the low bioavailability of the most promising natural compounds has determined the low effectiveness of the treatments: curcuminoids—0.47% [230], pterostilbene—35–80% [231], and resveratrol—20% [232]. PCA is produced following the ingestion of an anthocyanin-rich diet and is distributed through blood circulation to body tissues, where it remains longer than its parent compound [233]. The study of the anthocyanin bioavailability after ingestion and metabolism in humans stated that C3G, Pg-3-glc, P3G and their metabolite, PCA, were found in circulating blood [233,234]. Most studies have reported that the use of resveratrol via oral supplementation is limited by its poor solubility, rapid metabolism and low bioavailability [235]. Resveratrol is absorbed through epithelial diffusion in the gastrointestinal tract, forms complexes with transporter proteins, and is rapidly excreted from the body through the urinary tract. The half-life of resveratrol in the plasma of human volunteers has been reported to be 9.2–0.6 h after an oral dose of 25 mg [235]. Moreover, the local injection of resveratrol can overcome its low oral bioavailability and rapid first-pass metabolism [95]. To improve the low aqueous solubility of resveratrol, various methodological approaches, including liposomes, nanoparticles, and micelles, can be used to reduce the excretion speed and thus increase bioavailability. Moreover, dysbiosis is another critical factor that can alter the bioavailability of the site of interest. PPs have a low oral bioavailability, mainly due to extensive biotransformation mediated by phase I and phase II reactions in enterocytes and liver, but also by gut microbiota. Total PPs absorption in the small intestine is relatively low, being mainly bio-transformed by gut microbiota, followed by absorption in the bloodstream. Moreover, some PPs are not absorbed completely through the enterocyte due to their high hydrophilicity and molecular weight.

A major limitation of IA drug injection is that free drugs are cleared from the joint cavity rapidly, resulting in reduced drug bioavailability and the inconvenience of frequent injections. Moreover, some drugs that are highly insoluble in aqueous media form a crystal suspension that introduces the risk of crystal deposition. Furthermore, there is great variability in PP response due to intra-individual differences related to metabolic state, gut microbiota composition, and genetic profile [236]. Therefore, the stratification of patients according to their metabolic and lipidomic profiles is mandatory for improving efficacy [237].

Fourth, the complex and heterogeneous framework of OA disorder is another key aspect that has limited the application in the clinics of several promising molecules, described as the main reason for the failure of phase three clinical trials. In the clinic, OA is classified using the Kellgren-Lawrence (KL) classification [238], which identifies five grades of OA, ranging between 0 (no signs of OA) and 4 (severe OA). Usually, OA is defined as present starting from grade 2 upwards, however, the clinical presentation can be extremely heterogeneous with difficulties in selecting the most effective therapy.

Fifth, the lack of a universal regulatory framework among countries is critical to ensure product quality and reproducible results worldwide. This generates disorganization, with a lack of conformity to a global standard. The harmonization of the regulatory framework for BDMs across countries is mandatory to ensure better translation and to achieve global standards [239]. In contrast to in the US, in Europe, the European Food Safety Authority (EFSA) has outlined rules to ensure their pre-market safety [240] (Figure 3).

Finally, to better target new BDM-based strategies for OA it is crucial to further elucidate their molecular mechanisms beyond the BDM effect.

### 5.3. From Knowledge Gaps to New Opportunities: Perspectives of PPs in OA

Considering the current shortcomings in the biology of PPs, several perspectives can be envisaged to improve their safety and efficacy profiles, thanks to the increasing knowledge in basic and clinical research, and technological advances [6,8,241,242]. Most studies focused on the safety profiles of PPs on various cell systems in different formulations and with different delivery systems are necessary to provide more evidence-based medicine for the management of OA. Given the limited availability of different PPs compounds, there is an urgent medical need to develop novel delivery systems to guarantee their prolonged activity in the joint space, overcoming gastrointestinal side effects.

To tackle this challenge, a bulk of research has focused on the development of advanced nanoencapsulation and nanofabricated systems as suitable tools to deliver BDMs into the articular joint via IA delivery. Common bio-based nano-delivery systems for PPs primarily include protein-based systems, polysaccharide-based systems, and lipid-based systems [243]. These systems can include different formulations: liposomes (which efficiently entrap hydrophobic molecules), nanocrystals (which control the delivery of poorly water-soluble molecules), and nanoparticles based on polymers such as polylactic acid (PLA), lipids, or metals. The nano-encapsulation of PPs with different carriers can be a valid strategy to boost their bioavailability and efficiency [244]. Studies on DMM and MIA-induced OA models have proven that quercetin-loaded nanoparticle gel counteracts the typical degenerative OA features, such as proteoglycan degradation and the up-regulation of MMP-9, MMP-13, and ADAMTS-5 [208,209]. Recently, Wang et al. demonstrated that incorporating quercetin into nano-octahedral ceria allows the polarization of M1 toward the anti-inflammatory phenotype, M2, with important implications for the treatment of pain in OA [106]. Along this line, further studies have demonstrated the benefits of an IA injection of quercetin-loaded polycaprolactone microspheres to reduce the degenerative effects in a rat model, ensuring controlled release for over 30 days [245]. Concerning formononetin (FMN), which belongs to the 7-hydroxyisoflavones class, the formulation of a poly (ethylene glycol) (PEG)-formononetin (FMN) nano drug showed longer joint permanence and better anti-inflammatory effects than FMN alone [204]. All of the above-reported studies have proven that the use of new technological approaches can overcome three main gaps in the systemic administration of PPs: (1) minimizing safety risks; (2) improving the low bioavailability due to their rapid absorption, metabolization and excretion; (3) enhancing their healing effects. Further research studies are mandatory to guarantee the safety profile of the nanofabricated delivery systems, as their use in food applications remains scarce. Regarding the regulatory aspects of nanomaterials in the medical field, there are no specific rules to guarantee the safety of encapsulated products worldwide [241]. Moreover, the sterilization processes of these systems must be considered to ensure the chemical stability of PPs and their safety in humans, which is essential for the clinical use of drugs [244]. Given the low bioavailability of several PPs, there is a high medical need to develop BDM delivery systems to avoid the burst release and rapid clearance of BDMs in the joint space. Furthermore, this might be useful in the case of some BDMs with known side effects, e.g., gastrointestinal adverse effects of vanillic acid in vivo. To tackle this challenge, several approaches can be considered that have been used to promote IA drug delivery [246]: liposomes, nanocrystals, nanoparticles, lipids or metal, and micro-encapsulation by spray. Moreover, there is an urgent need for adequate and controlled release to ensure lasting effects. The release of quercetin from MPEG-PA hydrogel could be sustained for over 28 days, which means approximately 30% of quercetin remained in the hydrogel on day 28 [149]. Quercetin was encapsulated in polycaprolactone by the solvent evaporation method; quercetin release showed a biphasic nature due to the initial burst effect, followed by a controlled release [245]. Sheu et al. obtained an injectable hydrogel based on oxidized hyaluronic acid (oxi-HA) and resveratrol. First, sodium periodate was used to create the oxi-HA, whose functional group was further crosslinked with resveratrol [154]. Then, articular mouse chondrocytes were cultured within the gel and confirmed its good viability, as well as its good potential for reducing inflammatory reaction and damage [154,247]. Xiong et. al. demonstrated that metal–organic frameworks (MOFs) are materials with dense and large pores and abundant metal sites, with pH-responsive properties but poor water stability. They combined the most used MOFs, MIL-100, with HA, to improve its stability and dispersibility; they then combined it with protocatechuic acid to obtain MOF@HA@PCA. The protocatechuic acid was released in small amounts at pH 7.4, while the release amount increased with the reduction in the pH value (pH 5.6). It was released quickly in the first few hours, then there was a sustained and controlled release that reached a maximum within 24 h [157].

Notably, Ouyang et al. developed a delivery system capable of delivering the targeted release of hesperetin to chondrocytes, by using nanoparticles modified with cartilage affinity peptide (CAP, DWRVIIPPRPSA). They showed the cartilage-binding ability of nanoparticles that smartly released hesperetin, attenuating chondrocyte apoptosis and inflammation and articular cartilage degeneration [109].

Another innovative aspect to be studied is the coupling of BDMs with existing anti-inflammatory drugs, such as NSAIDs, or with other classes of BDMs. As already mentioned, this aims to reduce the dosage or number of drug administrations and, consequently, their side effects. PPs have been shown to suppress inflammation by inhibiting inflammatory cytokines and other catabolic and ROS mediators, while NSAIDs inhibit the pro-inflammatory enzyme COX. Considering their different mechanisms of action, their combination may be useful to enhance their healing potential. In this regard, the synergic effect of ferulic acid with other vitamins or active ingredients has provided better results, performance, and stability of the compounds [248]. Similarly, the co-injection of kaempferol and apigenin seems to increase efficacy [197]. Despite these promising findings, further studies are needed to examine the use of different classes of BDMs that could simultaneously target multiple cellular signaling pathways. Indeed, better-designed preclinical studies and larger clinical studies, using rigorous design and controls, are needed to reach clear-cut conclusions [249].

Another critical aspect to be taken into consideration for preclinical and clinical studies is the stratification of patients to minimize the issues related to heterogeneity in OA. In particular, an initial screening of the patient’s genetic, lipid, and metabolic pattern is of paramount importance to select which natural compounds may be most effective for personalized therapy. Unravelling the emerging factors influencing susceptibility/resistance to inflammation (genetic background, microbiota composition, metabolomic profiles) will be crucial to provide precise nutritional and lifestyle recommendations for specific groups of OA sufferers (e.g., personalized approaches based on inflammatory epigenetic signatures) [249,250]. In this context, the gut is a new and intriguing target for OA, as gut dysbiosis is one of the factors contributing to the pathophysiology of OA [251,252], where PPs have important roles. Lan et al. showed that the daily intragastric administration of quercetin in the MIA-induced OA model, from day 1 to day 28, partially reversed intestinal flora disorder [253]. Moreover, a more controlled and uniform regulatory management, by different countries, could help to reduce the large variability observed between different research and clinical studies worldwide, which makes it difficult to obtain clear-cut results and dietary recommendations.

In general, the challenges in this field are varied and, certainly, the interaction between a multidisciplinary team is necessary to speed up the development of future treatments for the management of OA (Figure 4).

## 6. Conclusions

In conclusion, PPs may have great potential for the treatment of OA due to their anti-inflammatory and anti-nociceptive properties. Several preclinical studies have reported how these natural compounds interfere with various inflammation and pain pathways and inhibit the release of inflammatory mediators, as well as molecules involved in matrix degradation. Results from clinical studies, although still limited in number, report the benefits of some compounds, such as resveratrol and curcumin, thus providing preliminary indications of their therapeutic potential. More recent advances in tissue engineering may allow the development of promising new formulations with greater stability and bioavailability at the damaged site of interest. Certainly, more and better-designed clinical trials, together with integrative transcriptomic, lipidomic, and metabolomic analyses, might provide a basis for precision treatments with PPs in OA patients.

## Figures and Tables

**Figure 1 ijms-23-15861-f001:**
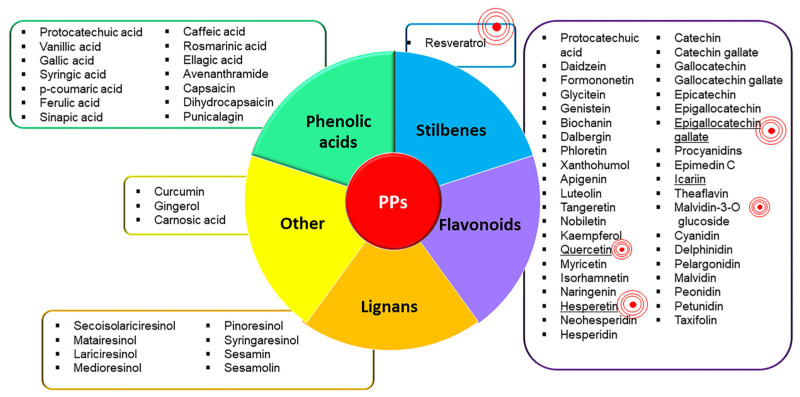
PPs classification. Figure 1 shows the known classes of PPs and the main PPs which constitute each class. The two classes of PPs most represented are flavonoids and phenolic acids. The ‘red concentric symbol’ highlights the most studied PPs in the field of OA inflammation and pain; its size corresponds to the number of studies. Overall, the most studied is resveratrol, followed by epigallocatechin gallate, quercetin, icariin, and hesperetin. To the best of our knowledge the following PPs have never been investigated regarding OA inflammation and pain: syringic acid, cryptochlorogenic acid, neochlorogenic acid, glycitein, dalbergin, neohesperidin, taxifolin, catechin, catechin gallate, gallocatechin, gallocatechin gallate, epicatechin, epigallocatechin, pelargonidin, peonidin, petunidin, dihydrocapsaicin, secoisolariciresinol, matairesinol, lariciresinol, medioresinol, syringaresinol, sesamolin.

**Figure 2 ijms-23-15861-f002:**
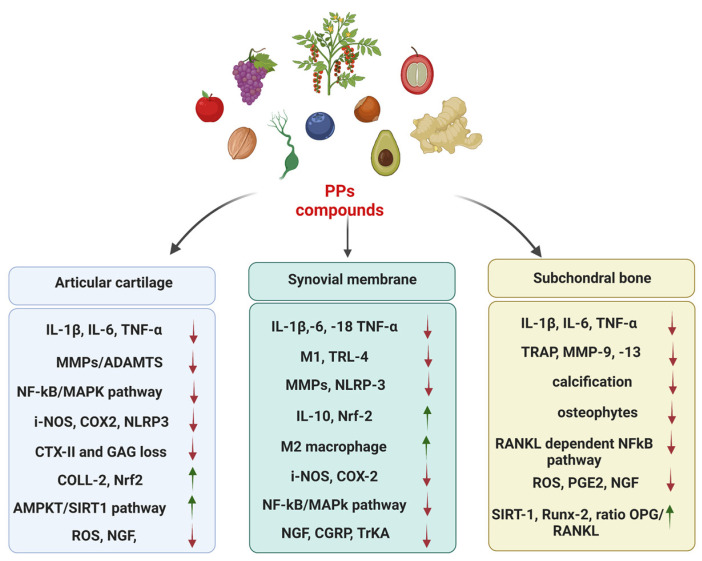
Overview and synthesis of the main biological effects of PPs compounds, reported in Table 1 and Table 2, on articular cartilage, synovial membrane and subchondral bone. Red arrows indicate down-regulation. Green arrows report up-regulation.

**Figure 3 ijms-23-15861-f003:**
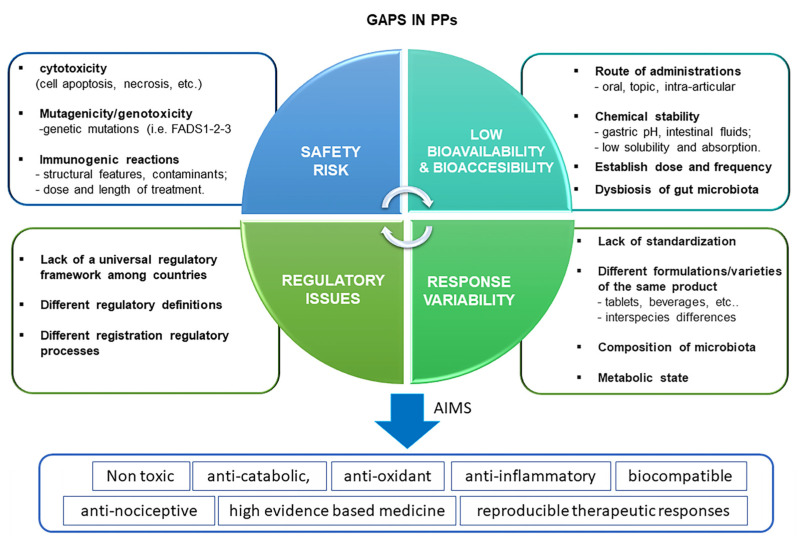
Schematic representation of main knowledge gaps for the use of PPs for the treatment of OA. The main limitations of their use include mainly the consideration of four aspects: safety, bioavailability, response variability and regulatory aspects. A safety profile should be devoid of cytotoxicity (absence of cell apoptosis, death, etc.), mutagenicity and genotoxicity and immunogenic reactions at a local and systemic level. Bioavailability depends on a certain number of factors, including the route of administration, chemical stability, dose and frequency of BDMs, and gut microbiota (dysbiosis). Response variability of BDMs is closely dependent on the lack of standardization, different formulations batch variations, metabolic state, and microbiota composition which render unreproducible therapeutic effects among OA patients. Finally, the regulatory aspect is another critical limitation due to the lack of universal rules which govern the BDMs process worldwide.

**Figure 4 ijms-23-15861-f004:**
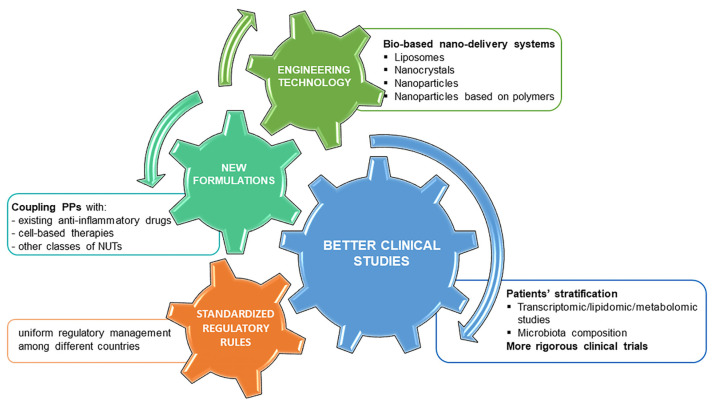
Overview of main perspectives to improve future research approaches studying PPs. The main challenges in this research field are addressed to improve their safety and efficacy profile. We distinguished four main research areas, engineering technology, new formulations, better clinical studies and standardized regulatory rules, which are closely interconnected and require different expertise. Engineering technology can represent a valid tool thanks to the new bio-based delivery systems to preserve PPs efficacy and ensure long-lasting effects in the joint. Better clinical studies should include previous patients’ stratification and more rigorous clinical trials. Combinatorial therapies with cell-based therapies, other BDMs or anti-inflammatory drugs can have synergistic benefits in reducing inflammation and pain. Finally, more standardized regulatory rules among countries could facilitate to reach and speed up of clear-cut conclusions on BDMs.

**Table 1 ijms-23-15861-t001:** PPs effects on inflammation and pain: in vitro models of OA.

Molecule Tested	In Vitro Models	Doses	Main Effects	Specific Outcomes	Ref.
Genistein	IL-1β-treated human OA chondrocytes	10 μM	Anti-inflammatory effect by stimulating Nrf2/HO1 signalling pathway	▪ ↓ NO, COX-2, MMP-9, MMP-1, MMP-3, MMP-13 ▪ ↑ HO-1 ▪ ↑ collagen II, aggrecan	[25]
6-gingerol	IL-1β-treated human OA chondrocytes	1–2–5–10 μM	Anti-inflammatory and anti-oxidant effects due to NRF2/GSTA4-4 pathway	▪ ↑ NRF2 phosphorylation and expression ▪ ↓ NO, PGE2, MMP-13, HNE ▪ ↑ GSTA4-4 ▪ No effect on p38 MAPK, JNK1/2, ERK1/2, ▪ p65-NF-kB phosphorylation	[26]
Resveratrol	IL-1β-treated human OA chondrocytes	6.25–25–50–200 μM	Anti-inflammatory effect by targeting IL-1β dependent activation of TLR4/MyD88/NF-κB signaling pathway	▪ ↓ TLR4 in chondrocytes ▪ ↓ TNF-α in supernatants ▪ ↓ NF-κB ▪ ↓ NF-κB nuclear translocation ▪ ↓ mature IL1-β ▪ ↓ MyD88, TRAF-6	[88]
Resveratrol	IL-1β-treated human OA chondrocytes	6.25–12.5–25–50–100–200 μM	Inhibition of inflammation	▪ ↓ MMP-13, IL-6 ▪ ↓ MyD88, p-IRAK4, TRAF6 (TLR4/MyD88-dependent pathway targets) ▪ ↓ TRIF (MyD88-independent pathway target)	[89]
Resveratrol	IL-1β-treated human chondrosarcoma cell line SW1353	50 μM	Anti-inflammatory effects by inhibition of TLR4/NF-κB signaling pathway	▪ ↑ PI3K/Akt ▪ inactivation of FoxO1 ▪ inactivating NF-κB ▪ inhibition of TLR4/MyD88—dependent and—independent signaling pathway	[90]
Resveratrol	Methylglyoxal-modified albumin (AGEs)-treated porcine chondrocytes	Pretreatment, 25–50–75–100 µg	Anti-inflammatory effect by inhibition of IKK-NF-KB and MAPK/AP- 1 signalling pathways	▪ ↓ iNOS, COX-2, NO, PGE2 ▪ ↓ IKKα/β-IκBα-NF-κB,AP-1, JNK DNA-binding activity ▪ ↑ IκBα ▪ ↓ IKKα/β, ERK phosphorylation ▪ ↓ MMP-13 activity ▪ ↑ collagen II ▪ ↓ proteoglycan release	[91]
Resveratrol	IL-1β-treated human chondrocytes	1–200 μM	Anti-inflammatory, anti-oxidative, and anti-apoptotic effects	▪ ↓ amount of mature IL-1β ▪ ↓ membrane-bound IL-1β ▪ ↓ caspase-3, cleavage of PARP, ▪ ↓ ROS ▪ ↑ ubiquitin-independent degradation of p53 ▪ ↓ p53-dependent apoptosis	[92]
Resveratrol	Normal and OA human chondrocytes	1–10–25–50 µM	Modulation of epigenome (SIRT modulator)	↑ SIRT1	[93]
Resveratrol	IL-1β-treated human OA chondrocytes	50 μM	ECM metabolism, autophagy, inflammation, apoptosis	↑ SIRT1/FOXO1 signaling	[94]
Resveratrol	IL-1β-treated mouse OA chondrocytes	2.3–23 µg/ml	SIRT1-dependent inhibition of inflammatory NF-kB-signaling	▪ ↑ SIRT1 ▪ ↓ phospho-p65 NFkB, HIF-2α, iNOS, MMP13 ▪ ↑ COL2A1, ACAN	[95]
Resveratrol	IL-1β-treated human OA chondrocytes	Pretreatment, 25 μM	Modulator of epigenome	▪ ↑ LINC00654 expression ▪ ↓ miR-210-5p expression ▪ ↑ OGFRL1 expression	[96]
Resveratrol	Coculture model of human chondrocytes and macrophages	10–25–50–100 μM	Inhibits the inflammatory amplification loop (IL-1β induced NF-κB and IL-6 in chondrocytes; IL-6 secreted activate STAT3 in macrophages; STAT3 positively regulate IL-6 secretion in chondrocytes).	▪ ↓ NFκB in chondrocyte ▪ ↓ STAT3 in macrophages	[97]
Curcuminoids Resveratrol Quercetin Pterostilbene Hydroxypterostilbene	Equine lymphocytes	10–20–40–80–160–320 μM	Decreased lymphocyte production of pro-inflammatory cytokines	▪ ↓ % of IFN-γ + and TNF-α + lymphocytes	[98]
Epigallocatechin 3-gallate	IL-1β treated chondrocytes	20–50 μM	Anti-inflammatory effect by targeting PTEN/miRNA-29b pathway	▪ ↓ miR-29b-3p ▪ ↓ MMP-13, IL-6 ▪ ↓ apoptosis	[99]
Epigallocatechin-3-gallate	IL-1β -treated human chondrocytes	10 to 100 uM	Anti-inflammatory response by inhibiting NF-κB and JNK-MAPK	▪ ↓ IL-1β ▪ ↓ TRAF-6	[100]
Epigallocatechin-3-gallate	IL-1β-treated human OA chondrocytes	Pretreatment, 20–50 μM	Anti-inflammatory effects	▪ ↑ miRNA-199a-3p ▪ ↓COX-2, PGE2	[101]
Epigallocatechin-3-gallate	IL-1β-treated human OA chondrocytes	100 μM	Inhibiting JNK	▪ ↓ JNK, JUN phosphorylation ▪ ↓ AP-1 binding activity No effect on p38-MAPK and ERKp44/p42 phosphorylation	[102]
Epigallocatechin-3-gallate	Human fibroblast-like synoviocytes and THP-1 cells treated with CPP crystals in presence of methyl-β-cyclodextrin	1–5–10 μM	Reduce the cytokine release induced by CPP crystals	▪ ↓ IL-1β, TNF-α, IL-8, CCL2 release ▪ ↓ chemokinetic of neutrophil and mononuclear cells	[103]
Quercetin	Tert-butyl hydroperoxide (TBHP)-treated rat chondrocytes	15–25–50–75–100 μM	Attenuate oxidative stress, ER stress, and associated apoptosis by activation of SIRT1/AMPK signalling	▪ ↓ cleaved caspase3, cleaved PARP ▪ ↑Bcl-2 ▪ ↓ ROS, MDA ▪ ↓ GRP78, CHOP ▪ ↑ SIRT1 ▪ ↓PERK, IRE1α phosphorylation ▪ ↓ATF6 ▪ ↑ AMPK phosphorylation	[104]
Quercetin	IL-1β-treated rat chondrocytes	Pretreatment, 8 µM	Anti-inflammatory and anti-apoptotic effects by suppressing IRAK1/NLRP3 signalling	▪ ↓ IL-18, TNF-α ▪ ↓ IRAK1, NLRP3, iNOS, COX-2,caspase 3 ▪ ↓ ROS ▪ ↓ apoptosis	[105]
Quercetin	IL-1β-treated articular cartilage human cells	10–100 µM	Anti-inflammatory and anti-apoptotic effect by inhibiting p38 MAPK signalling pathway	▪ ↓ IL-1β,TNF-α ▪ ↓ apoptosis ▪ ↓MMP-13, p38, ADAMTS-4 ▪ ↓ p38 phosphorylation ▪ ↑ collagen II	[106]
Quercetin	IL-1β-treated rat chondrocytes	2–4–8 μM	Anti-inflammatory, anti-catabolic, anti-apoptotic effect due to suppression of the Akt/NF-κB signalling pathway	▪ ↑ collagen II, aggrecan, GAG ▪ ↓ PGE2, NO, ADAMTS4, iNOS, COX-2, MMP-13 ▪ ↓ P65, AKT phosphorylation ▪ ↓ P65 nuclear translocation ▪ ↑ IkBα ▪ ↓ ROS ▪ ↓ caspase-3, Bax ▪ ↑ Bcl-2	[107]
Quercetin	RAW 264.7 cells	8 μM	Promote M2 polarization and pro-chondrogenic microenvironment for chondrocytes	▪ ↑ Arg, MR, Ym1 (M2) ▪ TNF-α, iNOS, IL-1β (M1) unchanged ▪ ↑ TGF-β1, TGF-β2, TGF-β3, IGF1 and IGF2 (pro-chondrogenic cytokines) ▪ ↑ STAT6, AKT phosphorylation	[107]
Quercetin	Coculture of RAW 264.7 cells and chondrocytes	8 μM	Indirect chondrogenic effect mediated by monocytes	▪ ↑ GAG, collagen II, sox9, aggrecan in chondrocytes	[107]
Hesperetin	IL-1β-treated chondrocytes		Anti-inflammatory effect by inhibition of NF-kB pathway	▪ ↓ iNOS, COX-2, NO, TNF-α, PGE2, IL-6 ▪ ↓ MMP-13, ADAMTS-5 ▪ ↓ NF-κB ▪ ↑ Nrf2	[108]
Hesperetin	IL-1β-treated mice chondrocytes	Loaded in Gd_2_(CO_3_)_3_-based nanoparticles (NPs)	Anti-inflammatory and anti-apoptotic effects by inhibiting TLR-2/NF-κB/Akt signalling	▪ ↓ Bax ▪ ↑ Bcl-2 ▪ ↓ IL-6, iNOS, TNF-α, COX-2 ▪ ↓ MMP-13, collagen X, MMP-12 ▪ ↑ proteoglycans, GAG ▪ ↑ aggrecan, sox-9, collagen II ▪ ↓ IKKα/β, IκBα, AKT phosphorylation ▪ ↓ TLR-2	[109]
Hesperidin	IL-1β-treated human OA chondrocytes	2.0 mg/ml	Anti-inflammatory effect of NF-κB inhibition	▪ ↓ IL-1β, IL-17A ▪ ↑ IL-6, IL-10 ▪ ↓ NO, PGE2, iNOS, COX-2, MMP-3, MMP-13 ▪ ↓ NF-κB activity	[110]
Icariin	LPS-treated rat chondrocytes	5–10–20 μM	Anti-inflammatory effect by inhibiting NLRP3 inflammasome-mediated caspase-1 signalling	▪ ↓ IL1-β, IL-18 ▪ ↓ MMP-1, MMP-13 ▪ ↑ collagen II ▪ ↓ caspase-1, ASC, NLRP3, GSDMD	[111]
Icariin	LPS-treated rat chondrocytes	20 μM	Anti-inflammatory effect by inhibition of TLR4/Myd88/NF-κB pathway	▪ ↓ TLR4, NF-κB p65, MMP-13 positive cells ▪ ↓ TLR4, Myd88, TRAF-6, MMP-3 ▪ NF-κB p65, IKK-α, IKK-β ▪ ↑ collagen II	[112]
Icariin	IL-1β-treated human SW1353 chondrosarcoma cells	20 μM	Anti-inflammatory effect by inhibition of p38/ERK/JNK	▪ ↓ MMP-1, MMP-3, MMP-13 ▪ ↓ p38, ERK, JNK phosphorylation	[114]
Icariin	LPS-treated synoviocytes	5–10 μM	Inhibition of ferroptosis during synovitis by the activation of the Xc-/GPX4 and NRF2 axis	▪ ↓ cell death ▪ ↓ MDA, iron ▪ ↑ GPX4, SLC7A11, SLC3A2L, NRF2 ▪ ↓ TFR1, NCOA4 ▪ ↓ RSL3	[116]
Carnosol (CL), carnosic acid (CA), carnosic acid-12-methylether (CAME), 20-deoxocarnosol and abieta-8,11,13-triene-11,12,20-triol (ABTT)	LPS-treated RAW264.7 cells	5–10–15 μM	Different anti-inflammatory potential	▪ ↓NO (CL, CA, CAME, 20-deoxocarnosol, ABTT similar effect), PGE2 (CL best effective) ▪ ↓ COX-1, COX-2 activity (CL, 20-deoxocarnosol, ABTT) ▪ ↓ iNOS, IL-1, IL-6, CXCL10/IP-10 (CL) ▪ ↓ IL-6, IL-1 (CAME) ▪ ↑ iNOS (ABTT) ▪ ↑ IL-6, MMP-9 (CAME, ABTT) ▪ ↑ COX-2 (CL, CAME)	[118]
Carnosol (CL)	IL-1β-treated chondrosarcoma SW1353 cells	6.25–12.5–25 μM	Anti-inflammatory and anti-catabolic effects	▪ ↓ iNOS, CCL5/RANTES, CXCL10/IP-10, MMP-13 ▪ ↑ aggrecan, collagen II, TIMP-1	[118]
Carnosol (CL), carnosic acid-12-methylether (CAME), abieta-8,11,13-triene-11,12,20-triol (ABTT)	IL-1β-treated human articular chondrocytes from knee	6.25–12.5 μM1.56–3.13 μM	Different anti-inflammatory potential	▪ ↓ CXCL8/IL-8, CCL20/MIP-3α, CCL5/RANTES, IL-1α, IL-1β, IL-6 (CL) ▪ ↓ MMP-3, MMP-13, ADAMTS-4, ADAMTS-5 (CL) ▪ ↑ IL-1α, IL-1β (CAME) ▪ ↓ CXCL10/IP-10, MMP-3, MMP-13, ADAMTS-4, LIF (CAME) ▪ ↓TNF-α, LIF, -5, MIP-3α, CCL5/RANTES, CXCL10/IP-10, IL-1α, -1β, -6, -8 ▪ ↓ p65 NFkB nuclear translocation (CL)	[118]
Caffeic acid phenethyl ester (CAPE)	IL-1β-treated human chondrocytes	10–20 μM	Anti-inflammatory effects via inactivation of NFkβ signalling pathway due to activation of Nrf2/HO-1 signalling pathway	▪ ↓ iNOS, COX-2, NO, PGE2 ▪ ↑ aggrecan, collagen II ▪ ↓ MMP3, MMP13, ADAMT5 ▪ ↓ IkBα, p65 phosphorylation ▪ ↑ IkBα ▪ ↓ p65 NFkβ nuclear translocation ▪ interacts at Keap1-NRF2 complex binding site (molecular docking analysis) ▪ ↑ Nrf2 nuclear translocation ▪ ↑ NRF2, HO-1	[119]
Tangeretin	IL-1β-treated mice chondrocytes	5–10–20 μM	Anti-inflammatory effect by blocking NF-κB by activating Nrf2 and MAPK pathway	▪ ↑ aggrecan, collagen-II ▪ ↓ MMP13, ADAMTS5, TNF-α, IL-6, iNOS, COX-2, NO, PGE2, ROS ▪ ↓ P65, ERK, JNK, p38 phosphorylation ▪ ↓ P65 NF-κB nuclear translocation ▪ ↑ IkBα ▪ ↑ Nrf2, HO-1 ▪ ↑ Nrf2 nuclear traslocation	[120]
Nobiletin	IL-1β-treated mice chondrocytes	Pretreatment, 10–20–40 μM	Anti-inflammatory and anti-catabolic effects by inhibition of NF-kB pathway	▪ ↓ NO, TNF-α, IL-6, PGE2 production and release ▪ ↓ COX-2, iNOS, MMP-3, -13, ADAMTS5 ▪ ↑ aggrecan, collagen II ▪ ↓p65 NFĸβ, IKKα/β, IKBα phosphorylation ▪ ↑ IKBα ▪ ↓ p65 NFĸβ nuclear translocation	[121]
Naringenin	IL-1β activated murine articular chondrocytes	20–40 µM	Anti-inflammatory effect by inhibition of NF-kB pathway	▪ ↓ MMP-3, -1, -13, ADAMTS-4, and ADAMTS-5 ▪ ↓ NF-kB p65, IkB-a phosphorylation ▪ ↓ MMP-3 activity	[122]
Theaflavin-3-3′-digallate	IL-1β-treated rat chondrocytes	20–40 μM	Anti-inflammatory effect by the PI3K/AKT/NF-κB and MAPK and Nrf2/HO-1 signaling pathway	▪ ↓ IL-6, TNF-α, iNOS, PGE2, COX-2 ▪ ↓ MMP13, MMP3, ADAMTS5 ▪ ↑ aggrecan, collagen II, SOX9 ▪ ↑ Nrf2, HO-1, SOD-2 ▪ ↑ Nrf2 nuclear translocation ▪ ↓ PI3K, AKT, p65 NFkB, ▪ IκBα, ERK, JNK, p38 ▪ ↓ p65 NFkB nuclear translocation ▪ ↑IκBα	[123]
Cyanidin	IL-1β-treated human OA chondrocytes	12.5–25–50–100 μM	Anti-inflammatory effect by inhibiting NF-κB pathway	▪ ↓ NO, PGE2, TNF-α, IL-6, iNOS, COX-2, ADAMTS5, MMP13 ▪ ↑ aggrecan, collagen II ▪ ↑ Sirt6 ▪ ↓ NF-κB pathway	[125]
Ellagic acid	IL-1β-treated human chondrocytes	50 μM	The anti-inflammatory effect through NF-κB inhibition	▪ ↓ iNOS, COX-2, NO, TNF-α, PGE2, IL-6, MMP-13, ADAMTS-5 ▪ ↑ collagen II, aggrecan ▪ ↑ IkBa ▪ ↓ p65 NF-κB nuclear translocation	[126]
Ferulic acid	IL-1β-treated human OA chondrocyte	1–5–10–20–20 μM	Anti-inflammatory effect by activating Sirt1/AMPK/PGC-1α signalling pathway.	▪ ↑ SIRT1, PGC-1α ▪ ↑ SIRT-1 activity ▪ ↑ AMPK phosphorylation ▪ ↓ IL-6, PGE2, nitrite, Collagen I, Runx-2, MMP-1, MMP-3, MMP-13 ▪ ↑Collagen II, Aggrecan ▪ ↓ ROS ▪ ↑ SOD-1, SOD-2, SOD activity	[127]
Fisetin	IL-1β-treated human articular chondrocytes	Pre-treatment, 1–5–10–25–50 μM	Anti-inflammatory effect of SIRT-1 activation	▪ ↑ SIRT-1 expression and activity ▪ ↓ NO, PGE2, IL-6, TNF-α ▪ ↓ iNOS, COX-2, MMP-3, MMP-13, ADAMTS-5 ▪ ↑ SOX-9, collagen II, aggrecan	[128]
Procyanidins	Chondrocytes		Attenuated apoptosis and senescence	▪ ↓ DPP4 ▪ ↑ SIRT-1	[129]
Curcumin	IL-1β-treated human chondrocytes	Exosomes derived from curcumin-treated MSC	Anti-inflammatory effect by targeting miR-124 and miR-143	▪ ↑ miR-124, miR-143 ▪ ↓ ROCK1 (target of miR-124), NFkB (target of miR-143) ▪ ↓TLR9 (effector of ROCK1)	[130]
Vanillic Acid	IL-1β/TNF-α-treated human OA chondrocytes	1 μM	Anti-inflammatory effect by targeting NF-κB signaling	▪ ↓CXCL12, CCL11, IL23A, MMP12, ADAMTS16, IL-6 ▪ ↑ COMP, GDF-5, TIMP-1, HMGB1 ▪ ↓INFγ, IL-1β, IL-2, IL-4, IL-12, IL-13, IL-6, IL-8, TNF-α ▪ ↓ IKKβ ▪ ↓ IκBα and p65 phosphorylation ▪ ↓MMP activity	[131]
Chlorogenic acid	IL-1β-treated rabbit chondrocytes	5–10–20 μM	Anti-inflammatory effect by inhibition of NF-κB	▪ ↓ MMP-1, MMP-3, MMP-13 ▪ ↑ TIMP-1 ▪ ↓ p65 NF-κB phosphorylation ▪ ↑ IκB-α	[132]
Biochanin	IL-1β-treated rabbit chondrocytes	5–25–50 μM	Anti-inflammatory effect via inhibition of IκBα/NF-κB activation	▪ ↓ MMP1, MMP3, MMP13 ▪ ↑ TIMP-1 ▪ ↓ P65 NF-κB ▪ ↑ IkBα	[133]
Sesamin	IL-1β- and/or OSM (long term) treated 3D human articular chondrocytes culture system	0.25–0.5–1 μM	Anti-inflammatory effect and anti-catabolic effect	▪ ↑ CAN, XT-1, XT-2, CHSY1, ChPF ▪ ↑ GAG synthesis and accumulation ▪ ↓ degraded GAG release ▪ ↑ DEC ▪ ↓ IL-1β endogenous expression and release	[135]
Sinapic acid	IL-1β-treated human OA chondrocytes	Pretreatment, 40–80–160–320 μM	Anti-inflammatory effects via activation of the Nrf2 signalling pathway	▪ ↓ IL-6, TNF-α, PGE2, NO ▪ ↑ Collagen II, Aggrecan ▪ ↓ MMP-9, MMP-13, iNOS, COX-2, disintegrin, ADAMTS-5 ▪ ↓ p65 NF-κB phosphorylation ▪ ↑ IκBα ▪ ↑ Nrf2/HO-1	[136]
Sinapic acid	IL-1β-treated mice OA chondrocytes	3–9–12 μM	Anti-inflammatory effects via activation of the Nrf2/HO-1 signalling pathway	▪ ↑ nuclear translocation of Nrf2 ▪ ↑HO-1 ▪ ↓ MMP-1, MMP-3, MMP-13, ADAMTS4, ADAMTS5	[137]
Xanthohumol	IL-1β-treated rat chondrocytes	10–25–50 μM	Anti-inflammatory effect by blocking the the activity of NF-κB via activating the Nrf2/HO-1 pathway	▪ ↓ iNOS, NO, TNF-α, IL-6, COX-2, MMP-13 ▪ ↑ Nrf2, HO-1 ▪ ↓ p65 NFkB translocation	[138]
Myricetin	IL-1β-treated human chondrocytes	5–10–15 μM	Anti-inflammatory effect by blocking NF-κB pathway and promoting PI3K/Akt mediated Nrf2/HO-1 axis	▪ ↓ NO, PGE2, TNF-α, IL-6, iNOS, COX-2, MMP13, ADAMTS5 ▪ ↑ aggrecan, collagen II ▪ ↓ P65 NF-κB nuclear translocation ▪ ↑ IkBα ▪ ↑ Nrf2 nuclear translocation, HO-1 ▪ ↑ Akt phosphorylation	[139]
Luteolin	Murine chondrocyte	20 µM	Anti-oxidant and anti-inflammatory effects by activating the AMPK/Nrf2 signalling pathway where AMPK acted as an upstream signal for Nrf2	▪ ↓ apoptosis, necrosis ▪ ↑ GSH/GSSG ▪ ↓ SOD, lipid peroxidation ▪ ↓ COX-2, iNOS, TNF-α, IL-6, NO, PGE2, ADAMTS5, MMP-13 ▪ ↑ aggrecan, collagen II ▪ ↑stabilization, nuclear translocation, and activation of Nrf2 ▪ ↑ HO-1, NQO1, GCLC (markers transcribed by Nrf2) ▪ ↑ AMPKα1 phosphorilation	[140]
Delphinidin	H_2_O_2_-treated human chondrocytes	40 μM	Anti-apoptotic effect by activating Nrf2 and NF-κB	▪ ↓ cleaved caspase-3, cleaved PARP ▪ ↑ Bcl-XL, Nrf2 ▪ ↑ NF-κB phosphorylated ▪ ↓ apoptosis ▪ ↑ autophagy (LC3)	[141]
Xanthohumol	IL-1β-treated rat chondrocytes	5–20 μM	Anti-inflammatory effect by activating HO-1 signalling	▪ ↓ NO, PGE2, TNFα, IL-6, MMP-3, MMP-13, ADAMTS-4, ADAMTS-5 ▪ ↑ aggrecan, collagen-II ▪ ↓ C/EBPβ and its translocation ▪ ↑ HO-1 ▪ ↑ interaction of HO-1 and C/EBPβ	[142]
▪ Protocatechuic acid (PCA) ▪ Cyanidin-3-O-glucoside (C3G) ▪ Pelargonidin-3-O-glucoside (Pg-3-glc) ▪ Peonidin-3-O-glucoside (P3G ▪ Malvidin-3-O-glucoside (M3G)	AGEs-induced human articular chondrocytes	Pretreatment, 2.5–5–10 μM	Anti-inflammatory effect by targeting NF-kappaB and MAPK signalling	↓ MMP-1, MMP-3, MMP-13 ↓ IKK, IκB, p65 phosphorylation ↓ ERK, p38, JNK phosphorylation C3G exhibits the highest inhibitory effect	[143]
Vanillic Acid	IL-1β-treated rat chondrocytes	5–10–20 μg/mL	Anti-inflammatory and anti hypertrophic effect by inhibition of MAPK and PI3K/AKT/NF-κB pathways	▪ ↓ COX-2, iNOS, MMP1, MMP3, MMP13, ADAMTS5 ▪ ↓ Collagen X, Runx-2, VEGF-A, HMGB1 ▪ ↑ Collagen II, aggrecan ▪ ↓ ERK, JNK, PI3K, AKT, p65, IκBα, Iκκα/β phosphorylation ▪ did not repress the p38 MAPK ▪ ↓ p65 NF-κB translocation	[125]
Rosmarinic acid	IL-1β-treated rat articular chondrocytes	Pre-treatment, 10–50–100 μM	Anti-inflammatory effect by NF-κB and MAPK pathway inhibition	▪ ↓ iNOS, COX-2, NO, PGE2, MMP-1, MMP-3, MMP-13 ▪ ↓ p38, JNK phosphorylation ▪ No effect on ERK phosphorylation ▪ ↓ p65 NF-κB translocation ▪ No effect on β-catenin	[144]
Fisetin	LPS-treated RAW264.7 mouse macrophages		Anti-inflammatory effect by JNK and NF-κB pathway inhibition	▪ ↓ NO, iNOS, COX-2, IL-6, TNF-α ▪ activation NF-κB ▪ ↑ JNK phosphorylation ▪ No effect on ERK and p38 MAPK	[145]
Punicalagin	LPS-treated rat chondrocytes	Pretreatment, 20–50–100 μM	Anti-inflammatory effect activating Foxo1 and downstream Prg4/HIF3α axis and autophagy	▪ ↓ IL-1β, TNF-a ▪ ↓ apoptosis ▪ ↑ Foxo1, Prg4, HIF3α, ULK-1, beclin-1, LC3, p- ULK1, p-Beclin1, LC3II/I, Foxo1 ▪ ↓ p62	[147]
Curcumin	IL-1β-treated human chondrocytes	▪ 100–200 μM ▪ Curcumin nanoparticles	Anti-inflammatory and chondroprotective effect	▪ ↓ IL-1β, TNF-α, MMP1, -3, -13, aggrecanase, ADAMTS5 ▪ ↑ CITED2	[151]
Resveratrol	Mice chondrocytes seeded on an oxidized hyaluronic acid hydrogel added with resveratrol	0.001 μM	Anti-inflammatory and anti-catabolic effects	↓ IL-1β, MMP-1, MMP-3, MMP-13 ↑ collagen II, aggrecan, sox-9	[154]
Resveratrol	LPS-treated Mice chondrocytes seeded on an oxidized hyaluronic acid hydrogel added with resveratrol	0.001 μM	Anti-inflammatory and anti-catabolic effects	↓ IL-1β, MMP-1, -3, -13 ↑ collagen II, aggrecan, sox-9	[154]
Resveratrol	IL-1β-treated human chondrocytes	100 μM	Anti-inflammatory and anti-apoptotic effects	▪ ↓ caspase-3, cleavage of PARP ▪ ↑ collagen II, β1-integrin	[155]
Resveratrol	IL-1β-treated human chondrocytes	100 μM	Anti-inflammatory and anti-apoptotic effects	▪ ↓ caspase-3, cleavage of PARP ▪ ↑ collagen type II, β1-integrin	[155]
Resveratrol	PBMC of healthy donors	1–5 μM	Increased inflammatory mediators	↑ IL-6 No changes in TNF-α, IL-8	[156]
Protocatechuic acid alone or loaded into a delivery system (MOF@HA@PCA)	IL-1β-treated OA chondrocytes	6 μM	Anti-inflammatory effect and anti-catabolic effect	▪ ↑ aggrecan, collagen II, GAG ▪ ↓ADAMTS5, COX2, IL-6, iNOS, MMP-1, MMP-3, MMP-13	[157]
Gallic acid	AGE-treated rabbit chondrocytes	Pretreatment	Prevented AGE-related changes	▪ ↑ ROS, SOD, collagen II, aggrecan ▪ ↓ iNOS, COX-2	[158]
p-Coumaric Acid (PCA)	IL-1β-treated rat chondrocytes		Anti-inflammatory effect and cellular senescence by targeting MAPK, NF-κB pathways	▪ ↓ COX2, iNOS, MMP1, MMP3, MMP13, ADAMTS4, ADAMTS5 ▪ ↑ collagen II, aggrecan ▪ ↓ 16INK4a phosphorylation, SAβ-gal activities	[159]
Ferulic acid	H_2_O_2_-stimulated porcine chondrocytes	Pretreatment, 40 μM	Inhibition of inflammatory response	▪ ↓ IL-1β, TNF-α, MMP-1, MMP-13 ▪ ↑ SOX9	[160]
Sinapic acid	IL-1β-treated rat chondrocytes		Anti-inflammatory effects via inhibition of the MAPK pathway	▪ ↓ NO, PGE2, iNOS, COX-2, MMP-1, MMP-3, MMP-13, disintegrin, ADAMTS-5 ▪ ↓ MAPK signalling	[161]
Caffeic acid	IL-1β-treated rat chondrocytes	5–10–20 μM	Anti-inflammatory effects via inactivation of NF-κB due to activation of the JNK pathway	▪ ↓ iNOS, COX-2 ▪ ↓ MMP-1, MMP-3, MMP-13, ADAMTS5 ▪ ↑ aggrecan, collagen II ▪ ↓ JNK, c-JUN, p65 phosphorylation ▪ no effect on p38, ERK phosphorylation ▪ ↓ p65 NF-κB nuclear translocation	[162]
Chlorogenic acid	IL-1β-treated human SW-1353 chondrocytes	31.25–62.5–125–250–500–1000 μM	Anti-inflammatory and effect via inhibition of IκBα/NF-κB activation	▪ ↓ iNOS, NO, IL-6, MMP-13, COX-2, PGE2 ▪ ↓ IkBα and p65 phosphorylation ▪ ↑ IkBα ▪ ↑ collagen II	[163]
Chlorogenic acid	IL-1β-treated human OA chondrocytes	5–10–20 μM	Anti-inflammatory effect inhibiting the iNOS-NO signalling pathway	▪ ↓ NO, PGE2, iNOS, COX-2	[164]
Formononetin	Human subchondral osteoblasts	0.01–0.1–1–10–100 μM	Inhibition of inflammatory mediators	▪ ↓ IL-6 ▪ ↓ ALP, VEGF, BMP-2, OCN, Col I, increased by OA	[165]
Genistein	IL-1β-treated OA chondrocytes	25–50–100 μM	Inhibition of inflammatory mediators	▪ ↓ TNF-α ▪ ↑ collagen II, aggrecan ▪ ↓ caspase 3 ▪ ↓ apoptosis	[166]
Biochanin	IL-1β-treated rat chondrocytes	5–7.5–10–15 μM	Anti-inflammatory, anti-catabolic, anti-oxidant responses via inhibition of NF-κB activation	▪ ↑ proteoglycan ▪ ↓ MMP-13, MMP-3, MMP-1, ADAMTS-5 ▪ ↓ iNOS, COX-2, PGE2 ▪ ↓TNFα, IL-6, IL-1α, IL-1β, INFγ, IL-2, GM-CSF, fractalkine, MCP-1, MIP-3α, LIX, and thymus chemokine-1, MMP-8, Fas ligand, ICAM-1, leptin, L-selectin, B7-2/CD86 ▪ ↓ NFκB phosphorylation ▪ ↓ NFκB nuclear translocation	[167]
Phloretin	IL-1β-treated human OA chondrocytes	10–30–100 μM	Attenuates inflammatory responses via inhibition of PI3K/Akt, NF-κB activation	▪ ↓ NO, PGE2, TNF-α, IL-6, COX-2, iNOS, MMP-3, MMP-13, ADAMTS-5 ▪ ↑ aggrecan, collagen-II ▪ ↓ PI3K/Akt phosphorylation ▪ ↓NF-κB activation	[168]
Apigenin	IL-1β-treated articular mice chondrocytes or IL-1β-treated cartilage explants	Pretreatment, 10–25–50 µM	Antioxidant and anti-inflammatory effect via Hif-2α regulation by JNK and NF-κB Pathways	▪ ↓ Hif-2α ▪ ↓ MMP3, MMP13, ADAMTS4, IL-6,COX-2 ▪ ↓ JNK phosphorylation and IκB degradation ▪ ↑ GAG, aggrecan in cartilage explants	[169]
Apigenin	IL-1β-treated rabbit articular chondrocytes	1–10–50–100 µM	Anti-catabolic effect	▪ ↓ MMP-1, -3, -13, ▪ ↓ADAMTS-4, -5 ▪ ↓ MMP-3 secretion ▪ ↓ MMP-3 activity	[170]
Luteolin	Cartilage cells of OA guinea pigs		Anti-inflammatory effect	▪ ↓ JNK, p38 MAPK ▪ ↓ NO, TNF-α, IL-6	[171]
Luteolin	IL-1β-treated rabbit articular chondrocytes	1–10–50–100 µM	Inhibited IL-1β-induced cartilage degradation	▪ ↓ MMP-3, MMP-1, MMP-13, ADAMTS-4, ADAMTS-5 ▪ ↑ collagen II ▪ ↓ MMP-3 secretion, activity	[172]
Kaempferol	Osteoarthritis chondrocytes		Decreased proinflammatory cytokine production and ECM degradation	▪ ↑ XIST/miR-130a ▪ ↓ STAT3	[173]
Kaempferol	IL-1β-treated rat chondrocytes	25–50–100–200 μM	Anti-inflammatory effect by inhibition of NF-kB pathway	▪ ↓ PGE2,NO ▪ ↓ iNOS, COX-2 ▪ ↓ p65, IkBα phosphorylation	[174]
Kaempferol	IL-1β-treated rat chondrocytes		Anti-inflammatory effect by inhibition of MApK-p38 pathway	▪ ↓ iNOS, COX-2 ▪ ↓ MMP-1,-3, -13, ADAMTS5 ▪ ↑ collagen II ▪ ↓ MAPK/p38	[175]
Epigallocatechin-3-gallate	Human chondrocytes		Anti-inflammatory effect	▪ ↓ COX-2, PGE2, IL-8	[176]
Isorhamnetin	IL-1β-treated human OA chondrocytes	Pretreatment, 10–50–100 µM	Anti-inflammatory effect by inhibition of NF-kB pathway	▪ ↓ MMP-3, MMP-13, iNOS, COX-2, NO, PGE2 ▪ ↓ NFkB p65 phosphorylation ▪ ↑ IκBα	[177]
Hesperetin	TNF-α-treated rat (SD) chondrocytes	1–5–10–20–50–100 μM	Anti-inflammatory effect	▪ ↓ IL-1β, PTGS2, MMP-13 ▪ ↓ degradation of the extracellular matrix ▪ ↑AMPK pathway	[178]
Hesperidin	H_2_O_2_-treated rat chondrocytes	Pretreatment, 0.1 µM	Anti-oxidant and anti-inflammatory effects	▪ ↓ MDA, ROS ▪ ↓ apoptosis ▪ ↑ collagen II, aggrecan, SOX9 ▪ ↓ caspase 3, IL-1β, TNFα, iNOS, MMP13	[179]
Icariin	IL-1β-treated rat chondrocytes		Anti-inflammatory effect by NF-κB pathway inhibition	▪ ↓ IL-6, TNF-α ▪ ↓ apoptosis ▪ ↓ p65 NFkB phosphorylation ▪ ↑ IKB-α	[180]
Icariin	Human osteoarthritis fibroblast-like synoviocytes (OA-FLSs)	0.1–0.5–1.5–10 μM	Anti-inflammatory effect	▪ ↓ IL-1β, MMP14, GRP78	[181]
Delphinidin	IL-1β-treated human chondrocytes	Pretreatment, 10–50–100 μM	Anti-inflammatory effect by inhibiting NF-κB pathway	▪ ↓ COX-2, PGE2 ▪ ADAMTS4, ADAMTS5 not modulated ▪ ↓ IRAK1, IKKα/β, NIK, IκBα phosphorylation ▪ ↓ IKKβ ▪ ↑ IκBα, IRAK ▪ ↓ p65 NF-κB nuclear translocation	[182]
Avenanthramide	IL-1β-treated human chondrocytes	50–100–200 μM	Anti-catabolic effect by modulating p38/JNK pathway	▪ ↓ MMP-3, -9, -12, 13, ADAMTS-4 ▪ ↓ p38,JNK phosphorylation ▪ degradation of IκB did not affect ERK, NF-κB, Ptgs2, Nos, collagen II, aggrecan, Sox9, degradation of IκB	[183]
Resveratrol	IL-1β-treated human OA chondrocytes	100 μM	Inhibits NFkB pathway	▪ ↓ NF-kB nuclear translocation ▪ ↓ NF-kB activation ▪ ↑ IkBa degradation	[184]
Curcumin Resveratrol	IL-1β-treated human OA chondrocytes	10 μM	Anti-apoptotic effect by inhibition of MAPK/ERK pathway	▪ ↓ apoptosis ▪ ↑ Erk1/2 ▪ ↓ caspase-3	[185]
▪ Resveratrol ▪ Curcumin	IL-1β-treated human OA chondrocytes	50 μM	Anti-inflammatory effects and anti-catabolic effect by inhibition of NF-κB	▪ ↓COX-2, MMP-3, MMP-9, VEGF ▪ ↓ NF-κB activation	[186]
Resveratrol	Adult human OA chondrocytes	10 μM	Anti-inflammatory effects	▪ ↓ CXCL1, IL-6, MMP-3, NOX4, PTGS2 ▪ ↑ HIF1A	[187]
Resveratrol	IL-1β-treated human OA chondrocytes	24–48 μM	Anti-inflammatory effects	▪ ↓ COX-2, MMP-1, MMP-3, MMP-13, iNOS ▪ ↑ collagen-II, aggrecan	[188]
Amurensin H (resveratrol dimer)	IL-1β-treated rat OA chondrocytes	4–8 μmol/L	Anti-inflammatory effects	▪ ↓ NO, iNOS ▪ ↓ PGE2, COX-2 ▪ ↓ IL-6, IL-17, TNF-α	[189]
Resveratrol	IL-1β-treated human chondrocytes	Pretreatment, 1–10 μM	Anti-inflammatory and anti-catabolic effects	▪ ↓ PGE2, LTB4 ▪ ↓ COX-2 activity ▪ ↓ proMMP-13, MMP-1, -3, -13 ▪ no effect on iNOS, NO	[190]
▪ Cyanidin-3-O-glucoside (C3G) ▪ Peonidin-3-O-glucoside (P3G) ▪ Protocatechuic acid (PA)	IL-1β-treated human chondrocytes	6.25–12.5–25–50 μM	Anti-catabolic effect by inhibiting NFĸB, ERK, MAPK pathway	▪ ↓ GAG, HA release (PA) ▪ ↓ MMP-1, MMP-3, MMP-13 ▪ ↑ IκBα ▪ ↓ JNK, IKK, p65, ERK, P38 phosphorylation (with differences among compounds)	[191]
Prodelphinidins	Human chondrocytes		Anti-inflammatory effect	▪ ↓ COX-2 ▪ ↓ PGE2	[192]
Epigallocatechin-3-gallate	AGE-BSA-treated human chondrocytes	Pretreatment, 25–50–100–200 μM	Reduced AGE-induced inflammation by inhibiting the MAPK and NF-kB activation	▪ ↓ TNF-α, MMP-13 expression and release ▪ ↓ p38-MAPK, JNK-MAPK, ERK-MAPK phosphorylation ▪ ↑ IκBα ▪ ↓ nuclear translocation ▪ of NF-κB p65 ▪ ↓ DNA binding activity of NF-κB p65 ▪ ↓IKKβ kinase activity	[193]
Epigallocatechin-3-gallate	IL-1β-treated human OA chondrocytes	1–100 μM	Inhibiting NF-kB/not through IKK but through the proteasome pathway	▪ ↓ NO, iNOS ▪ ↓ NFkB nuclear translocation ▪ No effect on IKK activity ▪ ↑ IκBα	[194]
Epigallocatechin-3-gallate	IL-1β-treated human chondrocytes	Pretreatment, 100–200 μM	Anti-inflammatory effect	▪ ↓ COX-2, iNOS, NO, PGE2 ▪ ↓ COX-2, iNOS activity	[195]
Carnosic acid (CA)	IL-1β-treated human chondrocytes	1–50 μM	Prevented IL-1β-dependent decrease in HO-1 by increasing miR-140	▪ ↑ HO-1 (also in un-treated human chondrocytes) ▪ ↑ miR140 targeting BACH-1 (HO-1 transcriptional inhibitor) ▪ ↓ MMP-13, ADAMTS-5	[196]
Carnosic acid (CA)	Mouse femoral head cartilage explants	1–50 μM	Suppressed proteoglycan release	▪ ↓ GAG release	[196]
Sesamin	IL-1β-treated porcine cartilage explant culture systems	0.25–0.5–1 μM	Anti-inflammatory and anti-catabolic effects	▪ ↓ HPR, sulphated-GAG release ▪ ↓ MMP-1, MMP-3, MMP-13 expression ▪ ↓ MMP-3 activity ▪ ↓ p38, JNK but not ERK1/2 phosphorylation ▪ Do not influence aggrecanase activit	[197]

**Table 2 ijms-23-15861-t002:** PPs effects on inflammation and pain: in vivo models of OA.

Molecule Tested	In Vivo OA Models	Doses and Duration Delivery Routes	Main Effects	Specific Outcomes	Ref.
Resveratrol	MIA-induced OA model in mice	10, 100 µg administration after 1 and 4 weeks from OA induction IA injections	Prevention of OA progression	▪ ↑ SIRT1 chondrocytes ▪ ↓ phospho-p65 NFkB, HIF-2α in chondrocytes ▪ ↓ iNOS, MMP-13 in cartilage ▪ ↑ COLL-2	[95]
Quercetin (QUE)	MMTL-(DMM) induced OA in rats	50–100 mg/kg QUE, once daily for 12 weeks Intraperitoneal injection	Reduced cartilage degeneration, apoptosis, ER stress	▪ ↓ OARSI score ▪ ↓ apoptosis in cartilage ▪ ↓ caspase-3 in cartilage ▪ ↓ 8-OHdG (oxidative stress marker) in cartilage ▪ ↑ SIRT1, p-AMPK in cartilage ▪ ↓ CHOP, ATF6, p-PERK, p-IRE1α in cartilage	[104]
Quercetin	ACLT-induced OA in rats	50–100 mg/kg once daily for 12 consecutive weeks Intraperitoneal injection	Anti-inflammatory and chondroprotective effects	▪ ↓ OARSI score ▪ ↓ IL1β, IL-18, TNF-α ▪ ↓ NLRP3, IRAK1 caspase 3: high dose	[105]
Quercetin	MMT + AMTL-induced OA in rats	8 μM once a week for 6 consecutive weeks IA injection	Anti-inflammatory and immunomodulatory effects	▪ ↓ OARSI score ▪ ↑ TGF-β1, TGF-β2 in the synovial fluid ▪ ↓ M1 and ↑ M2 in synovia ▪ ↓ MMP-13 in cartilage ▪ ↑ COLL-2, aggrecan in cartilage ▪ ↓ chondrocyte apoptosis	[107]
Hesperetin-Gd_2_(CO_3_)_3_@PDA-PEG-DWpeptide, (HGdPDW) Gd_2_(CO_3_)_3_-based nanoparticles (NPs)	ACLT-induced OA in mice	5% (*w/w*) 3 times/week for 8 weeks IA injection	Alleviation of cartilage degeneration and apoptosis	▪ ↓ TLR-2 ▪ ↓ OARSI score ▪ ↑ proteoglycans ▪ ↓ apoptosis of chondrocytes	[109]
Icariin	MIA-induced OA in rats	20 μM 2 weeks post-MIA injection IA injection	Protective effect on OA degradation by inhibition ofNLRP3-mediated pyroptosis	▪ ↓ cartilage erosion ▪ ↓ MMP-1,-3 NRLP3, IL-1β, and IL-18 in cartilage ▪ ↑ COLL-2	[111]
Icariin	KOA rat model established by the Hulth method	20 mg/kg, daily Intra-gastric administration	Protective effect on cartilage and subchondral bone degeneration; neuromodulation; analgesic effect	▪ ↓ cartilage degradation ▪ ↓ subchondral bone degeneration ▪ ↑ mechanical withdrawal threshold, thermal withdrawal latency ▪ ↓ IL-1β, TNF-α in serum ▪ ↓ MMP-3, MMP-13, TGF-β, BMP-4, SMAD-4 in cartilage ▪ ↓ SMAD 1/5/9 phosphorylation in cartilage ▪ ↓ NPY, NPY1R, SP R and 5-HT1B R, SP, CGRP in serum ▪ ↑ VIP in serum ▪ reversed changes in ReHo by MRI ▪ ↓ Ncam1 and Trpm2, Rtn4 in brain	[112]
Caffeic acid phenethyl ester (CAPE)	DMM model in rats	10 mg/kg/2 days for 8 weeks from DMM surgery Intraperitoneal injection	Delay of OA progression	▪ ↓ MMP-3, -13in cartilage ▪ ↑ COLL-2 in cartilage ▪ ↑ Nrf2 nuclear translocation in cartilage	[119]
Tangeretin	DMM-induced OA in mice	10–20 mg/kg daily intragastric dose	Anti-degenerative and anti-inflammatory effects by blocking NF-κB by activating Nrf2 and MAPK signalling	▪ ↓ OARSI scores ▪ ↑ aggrecan in cartilage ▪ ↓ MMP13 in cartilage ▪ ↓ p65 NF-κB, p38 phosphorylation in cartilage	[120]
Nobiletin	DMM-induced OA in mice	20 mg/kg every 2 days for 8 weeks Intraperitoneal injection	Delay of OA progression	▪ ↓ OARSI score ▪ ↓ synovitis ▪ ↓ MMP-13 in chondrocyte ▪ ↓ cartilage destruction	[121]
Naringenin	MIA-induced OA in rats	20–40 mg/kg daily, two weeks after OA induction Intragastric delivery	Pain alleviation	▪ ↓ pain behaviour (paw withdrawal latency and paw withdrawal threshold) ▪ ↓ MMP-3 in articular cartilage ▪ ↓ chondrocyte death, cartilage erosion and fibrillation (40 mg/kg)	[122]
Theaflavin-3,3′-Digallate	DMM model in rats	4 mM, 2 weeks after OA induction every 2 days for 6 weeks IA injection	Protection for cartilage degradation	▪ ↓ OARSI score ▪ ↓ cartilage degradation, GAG loss ▪ ↑ COL2, Nrf2 in cartilage	[123]
Ellagic acid	DMM mouse model	40 mg/kg intragastric administration, every 2 days for 8 weeks	Prevent the progression of OA degeneration and inflammation	▪ ↓ cartilage destruction, proteoglycan loss ▪ ↓ OARSI score ▪ ↓ synovitis	[126]
Fisetin	DMM-induced OA in mice	20 mg/kg daily for 8 weeks after surgery Oral gavage	Reduction in synovitis and joint destruction	▪ ↓ proteoglycan ▪ ↓ OARSI scores ▪ ↓ bone plate thickness ▪ ↓ synovitis	[128]
Curcumin	OA mouse model	exosomes derived from curcumin-treated MSCs	Inflammatory markers modulation	▪ ↓ apoptosis ▪ ↑ miR-124, miR-143 ▪ ↓ ROCK1, NFkB (target of miR-124) ▪ ↓TLR9 (effector of ROCK1)	[130]
Pinoresinol diglucoside	ACLT-induced OA in rabbits	2 mg/mL after 4 weeks from OA induction, once a week for 5 weeks IA injection	Cartilage-protecting effects	▪ ↓ TIMP, serum IL-1β, IL-6, and TNF-α levels, and PI3K and AKT activation; ▪ ↑ MMP-1 expression and Bcl2/Bax ratio	[134]
Sinapic acid	DMM model in mice	10 mg/kg/day from DMM surgery Intragastric delivery	Prevention of OA progression by reducing inflammation and catabolism	▪ ↓ TNF-α, IL-1β, IL-6 ▪ ↓ MMP-1, -3, -13 ▪ ↓ADAMTS-4, -5 ▪ ↑ Nrf2, HO-1	[137]
Xanthohumol	DMM model in mice	40 mg/kg XN in 0.5% carboxymethylcellulose (CMC) daily for 8 weeks ntragastric delivery	Safeguarding influence in OA advancement	▪ ↓ calcification ▪ ↓ osteophytes ▪ chondroprotection: ▪ ↑ GAG content and ▪ ↓ OARSI score ▪ ↑ Nrf2	[138]
Myricetin	DMM-induced OA in mice	20 mg/kg myricetin in 0.5% carboxymethylcellulose (CMC), every two days for 8 weeks Intragastric administration	Reduction in joint destruction	▪ ↑ proteoglycans and GAGs ▪ ↓ OARSI scores ▪ ↑ NRF2 translocation ▪ ↑ pAKT	[139]
Luteolin	DMM-induced OA in rats	10 mg/kg/day in 0.5% carboxymethylcellulose (CMC) or vehicle alone (0.5% CMC) for 8 consecutive weeks Intra-gastric delivery	Prevention of cartilage damage	▪ ↓ OARSI score ▪ ↓ MMP-13 in cartilage ▪ Nrf2 in cartilage	[140]
Punicalagin	ACLT-induced OA model in rats	10 mg/kg/day Oral gavage	Alleviates the inflammatory injury and ECM degradation through Foxo1/Prg4/HIF3α pathway activation and autophagy	▪ ↓ OARSI score ▪ ↓ cartilage destruction ▪ ↓ chondrocyte apoptosis ▪ ↓ TNF-α, IL-1β in synovium ▪ ↑ Foxo1, Prg4, HIF3α, p-ULK1, p-Beclin1 and LC3II/I ▪ ↑ Aggrecan, Collagen II ▪ ↓ ADAMTS5, MMP13, p62	[147]
Quercetin	MMX + ACLT + PCLT-induced OA in rabbits	25 mg/kg from the 5th week on, once daily for 4 weeks Intragastric delivery	Reduction in OA progression	▪ ↓ MMP-13 in SF and synovium ▪ ↑ SOD, TIMP-1 in SF and synovium ▪ ↓ cartilage degeneration	[148]
Quercetin encapsulated in the mPEG-polypeptide thermogel	ACLT-induced OA model in rats	50 or 500 μg IA injection	Relief of pain symptoms and delay of OA progression	prolonged effect of quercetin ↓ pain chondroprotection (↓ OARSI score)	[149]
Palmitoylethanolamide co-ultra micronized with quercetin (PEA-Q)	carrageenan paw oedema and MIA-induced OA models in rats	10 mg/kg three times per week for 4 weeks Oral administration	Reduction in inflammatory and pain responses	carrageenan paw oedema ↓ paw oedema ↓ inflammation ↓ thermal hyperalgesia MIA model ↓ cartilage degradation (↓ Mankin score) ↓ TNF-α, IL-1β ↓ MMP-1, -3, -9 ↓ NGF ↓ mechanical allodynia	[150]
Curcumin	MMTL-induced OA in mice	curcumin nanoparticles (0.07 mg of 10 μg curcumin/1 mg nanoparticles) once daily for 8 weeks Topic administration	Delay of OA progression and pain relief no significant effect on OA pain relief.	▪ ↓ pain ▪ ↓ MMP-13, ADAMTS-5 ▪ ↓ cartilage erosion ▪ ↓ synovitis ▪ ↓ adipokines and pro-inflammatory mediators	[151]
Next Generation Ultrasol Curcumin (NGUC)	MIA-induced OA in rats	100 mg/kg of NGUC (20 mg/kg of curcuminoids) 200 mg/kg of NGUC (40 mg/kg of curcuminoids) Oral administration for 4 weeks	Enhanced bioavailability of curcumin and amelioration of OA pathophysiology	▪ ↓ joint swelling ▪ ↓ TNF-α, IL-1β, IL-6, COMP, and CRP, MMP-3, 5-LOX, COX-2, and NFκB in synovium	[152]
Malvidin	MIA-induced OA in rats	10, 20 mg/kg/day, dissolved in saline (vehicle)] was administered for 14 days starting on the same day of MIA injection IA injection	Pain-relieving effects	▪ Antinociceptive and pain-relieving effects ▪ ↓ IL-1β, IL-6, TNF-α, and MMPs ▪ inhibition of the NF-κB pathway	[153]
PCA loaded into a delivery system (MOF@HA@PCA) pH-responsive nanoparticle system	ACLT-induced OA in rats	6 μg/mL(once/week) for 4 and 8 weeks from ACLT surgery IA injection	Attenuation of OA progression	▪ ↑ cartilage regeneration: macroscopic/OARSI score reduction ▪ ↓ MMP-13	[157]
Genistein	DMM + ACLT + MCLT-induced OA in rats	20 mg/kg/each day for 6 weeks Intragastric delivery	Suppression of inflammation, cartilage degradation and apoptosis	▪ ↓ TNF-α,IL-1β in synovial fluid ▪ ↑ collagen II, aggrecan, GAG in cartilage ▪ ↓ caspase 3 in cartilage	[166]
Apigenin	The IL1-β-treated knee joint of rats	50–100 μM pretreatment IA injection	Chondroprotection	▪ ↓ MMP-3 production	[170]
Luteolin	IL-1β-treated rabbit articular chondrocytes	50–100 μM pretreatment Articular injection	Chondroprotection	▪ ↓ MMP-3 production	[172]
Hesperidin	Medial collateral ligament transection + medial meniscus removal- induced OA model in rats	200 mg/kg body weight once 118 days for 28 days Oral administration	Delay of cartilage degeneration	▪ ↓ cartilage destruction ▪ ↓ IL-1β,TNF-α ▪ ↑ proteoglycans	[178]
Sesamin	Papain-induced OA model in rats	1,10 μM sesamin injection every 5 days for 25 days	Inhibit the pathological progression	▪ ↑ PGs, collagen II	[196]
Apigenin and synovial membrane-derived mesenchymal stem cells (SMMSCs)	ACLT-induced OA in rats	Apigenin (0.1–0.3 μM) ± SMMSC (3 × 106) once weekly for 3 weeks IA injection	Suppression of inflammation and oxidative stress supplementary benefits when combined with cell-based therapy	▪ ↓ IL-1β, TNF-α, MDA ▪ ↑ SOD ▪ ↑ SOX-9, COLL-2 and aggrecan	[198]
Apigenin and kaempferol ± SMMSCs	ACLT-induced OA in rats	kaempferol (10 or 20 μM) + apigenin (0.1 μM) weekly, for three weeks IA injection	Coupling the two molecules has synergistic benefits in reducing inflammation and cartilage degradation	▪ ↓ TNF-α, IL-1β and MDA ▪ ↓ MMP-3, -13, iNOS ▪ ↑ SOD ▪ ↑ SOX-9, COLL-2, aggrecan	[199]
Capsaicin	MIA-induced OA in rats	0.5% two weeks before OA induction single IA injection	Suppression of mechanical pain and bone erosion	▪ ↓ mechanical pain ▪ protective effects on bone microarchitecture	[200]
Epigallocatechin 3-gallate	Spontaneous OA in guinea pig	10 µM once a week for 12 consecutive weeks from 6 months of age IA Injection	Inhibition of age-related-OA changes	▪ ↓IL-1 β and COX-2 ▪ ↓ cartilage degradation (OARSI reduction) ▪ ↓ MMP-13 and p16 Ink4a	[201]
Epigallocatechin 3-gallate	ACLT-induced OA model in rats	10 μM once every three days for 5 weeks, 2 weeks after the operation IA Injection	Prevention of inflammation and the promotion of autophagy	▪ ↓ synovitis ▪ ↓ COX-2 and MMP-13 ▪ ↓ cartilage degradation (OARSI reduction) ▪ ↓ COLL-X ▪ ↑ COLL-2 ▪ ↑ Beclin-1 and LC3 (autophagic markers)	[202]
Genistein	MIA-induced OA in rats	30 mg/kg/day for 8 weeks Intragastric delivery	Prevention of cartilage damage	▪ ↓ MMP-8,-13, and IHH ▪ ↑ Sox5, and Sox6	[203]
Protocatechuic acid (PCA)	ACLT-induced OA in rats	50 mg/kg/day for 4 weeks from ACLT surgery Intragastric delivery	Suppression of osteoclastogenesis	▪ ↓ CTX–I/II ▪ ↓ c-Src and IL-6 ▪ ↓ MAPK, ATK and NF-κB pathways	[204]
Poly(ethylene glycol) (PEG)-formononetin (FMN) nanodrug	ACLT-induced OA in rats	1.25 μg/mL (once per week) for 4 or 8 weeks IA injection	Anti-inflammatory and chondroprotective effects	▪ chondroprotection (reduction OARSI score) ▪ ↑ COLL-2 ▪ ↓ MMP-13	[205]
Quercetin	Papain-induced OA in rats	1, 4 and 7 days post-quercetin treatment 5–10 mg/kg Intragastric delivery	Prevention of OA progression	▪ ↓ serum IL-1β and TNF-α ▪ ↓TLR-4 and NF-κB	[206]
Quercetin	MIA-induced OA model in mice	20 mg/kg for 28 days Intragastric delivery	Reduction in OA symptoms and cartilage degradation	▪ ↓ inflammation in the tibia and femur ▪ ↓ serum MMP-3 and MMP-13	[207]
Quercetin	MNX-induced OA model in rats	100 mg/kg on day 0 for consecutive 7 days once daily Oral administration	Relief of OA conditions	▪ ↓ ROS ▪ ↑ GSH, GPx ▪ ↓ NO, MMP-3, -13 ▪ ↑ AMPK/SIRT1 pathway	[208]
Quercetin-loaded lecithin-chitosan nanoparticles	DMM and MIA-induced OA models in rats	Dose 1 (0.84 mg/g gel), dose 2 (1.68 mg/g gel), dose 3 (3.36 mg/g gel) on the 29th day Topic administration	Prevention of cartilage degradation	▪ ↓ IL-1 β, ADAMTS-5 ▪ ↓ MMP-9,-13,	[209]
Resveratrol	DMM-induced OA model in rats	0.8 mL of 5, 10, and 15 μmol/L, 6 weeks after OA induction IA injection	Cartilage protection	▪ ↑ SIRT1 ▪ ↓ p53	[210]
Resveratrol	MIA-induced OA model in rats	5 or 10 mg/kg body/daily after OA induction for up to 14 days Oral administration	Inhibition of inflammatory mediators	▪ ↓ COX-2, i-NOS ▪ ↓ serum levels of IL- 1β, IL-10 and TNF-α ▪ ↓ IL-1β, IL-10, -6, MMP-13 and TNF-α mRNA ▪ comparable effect to etoricoxib, a reference drug.	[211]
Resveratrol	MIA-induced OA model in rats	50 μg/10 μL, 250 μg/10 μL, and 500 μg/10 μL for 8 consecutive days from day 14 today 21 after OA inductions Intrathecal administration	Reduction in mechanical allodynia	▪ ↑ SIRT1 ▪ ↓ p53 ▪ ↑ paw withdrawal threshold	[212]
Resveratrol	Type II collagenase-induced OA model in rats	40 and 80 mg/kg, twice daily for 4 weeks Intragastric administration	Reduction in joint destruction via anti-inflammatory and antiapoptotic effects	▪ ↓ serum levels of IL-1β, IL-6, TNF-α, and MCP-1 ▪ ↓ TLR-4,MyD88, and NF-κB p65 in articular cartilage ▪ ↓caspase-9 and Bax protein levels	[213]
Resveratrol	Obesity-related OA model: high-fat diet (HFD)-induced OA in mice	22.5 and 45 mg/kg for 12 weeks, after OA induction. Intragastric administration	Alleviation of OA pathology via the reduction in systemic inflammation	▪ ↓ serum IL-1β, leptin ▪ ↓TLR4, TRAF6 in cartilage	[214]
▪ Resveratrol-loaded TEMPO-oxidized cellulose aerogel ▪ Resveratrol	OA induced by treadmill-running exercise in rats	25 mg Resveratrol or RLTA (containing 25 mg Res), daily for 3 weeks Intra-gastric administration	Reduction in inflammation	▪ ↑ Sirt1 in articular cartilage ▪ ↓ P38, COX-2, MMP13 in articular cartilage ▪ ↓ IL-6, TNF-α in SF	[215]
Vanillic acid	MIA-induced OA in rats	30 mg/kg/day for 14 days from MIA Intragastric delivery	Reduction in synovitis and pain-related behaviour	▪ ↓ CGRP, NGF, and TrkA in synovium ▪ ↓ IL-1β and IL-18 ▪ ↓caspase-1, ASC, and NLRP3	[216]

**Table 3 ijms-23-15861-t003:** Clinical studies of PPs on OA patients.

Molecule Tested	Study Types	Patients Data	Treatments (Route-Doses)	Follow-Up	Main Effects	Ref.
**Curcumin (Cur) versus diclofenac (Df)**	Randomized open-label parallel-arm study	knee OA VAS score ≥ 4 *N* = 140 *N* = 71 Cur + Df (50 ♂ + 21♀) *N* = 69 Df (48 ♂ + 21♀) aged 38–65 year	Cur complex 500 mg + Df 50 mg (Cur + Df)/ 2 times daily for 28 days Df 50mg alone 2 times daily for 28 days Oral administration	Baseline, Day 14 Day 28	▪ Superior improvement in KOOS after the treatment with Cur + Df ▪ Minor adverse effects in the group treated with Cur + Df ▪ Reduced number of patients in the group treated with Cur + Df required additional rescue analgesics ▪ Greater improvement in pain and functional capacity in the group treated with Cur + Df	[217]
**Curcumin-phosphatidylcholine phytosome complex** **(Meriva^®^)**	clinical trial	knee OA mild to moderate pain *N* = 100 *N* = 50 NSAIDs *N* = 50 Meriva^®^	NSAIDs vs. Meriva^®^ Two 500 mg tablets daily (200 mg curcumin/day) Oral administration	Baseline, 8 months	▪ Meriva^®^ is an anti-inflammatory agent ▪ Improved pain sensation, joint stiffness, and physical function in the group treated with Meriva^®^ ▪ Reduction in inflammatory markers: SCD40L, IL-lß, IL-6, sVCAM-1, and ESR in the group treated with Meriva^®^ ▪ Decreased use of NSAIDs after Meriva^®^ treatment	[218]
**Resveratrol**	Open-labelednoncontrolled clinical trial	knee OA mild-to-moderate OA *N* = 28 (8 ♂ + 20♀) KOOS = 46 ± 13 VAS = 74 ± 8	500 mg/day in a single oral dose for 90 days (monotherapy) Oral administration	Baseline, day 30, 3 months	▪ ↑ Aggrecan ▪ ↓VAS score ▪ ↓ KOOS score ▪ IL-6, IL-1β,TNF-α- unchanged	[219]
**Resveratrol + meloxicam**	Randomized Placebo-Controlled Study	knee OA mild-to-moderate OA *N* = 110 VAS = 100 ▪ *N* = 60 Rsv + Mx (13 ♂ + 37♀) ▪ *N* = 50 Placebo + Mx (10 ♂ + 22♀)	15 mg/day meloxicam and either 500 mg/day Resveratrol or placebo for 90 days Oral administration	12 weeks 90-days	▪ Significant decrease in pain score in the group treated with Rsv ▪ Reduction in serum levels of IL-1β, TNF-α, PCR, complement proteins C3 and C4	[220]
**Combination of a glucosamine-chondroitin-quercetin glucoside (GCQG)**	Clinical study	Knee OA mild-to-moderate OA 62.5 ± 9.4 years *N* = 46 (7 ♂ + 39 ♀)	glucosamine hydrochloride (1200 mg/d), shark cartilage powder (300 mg/d) containing 75–111 mg of chondroitin, and 45 mg of quercetin Oral administration	3 months	▪ Improvement in the walking, stairs, JOA and VAS scores ▪ Reduction in the SF chondroitin-6 sulfate (CS6)	[221]
**Glucosamine hydrochloride, chondroitin sulfate and quercetin glycosides** **(GCQ supplement)**	A randomized double-blind, placebo-controlled study	Knee OA *N* = 40 *N* = 20 GCQ treatment (4 ♂ + 16♀) *N* = 20 placebo (3 ♂ + 17♀) KL grade I-II	1200 mg glucosamine hydrochloride, 60 mg chondroitin sulfate and 45 mg quercetin glycosides per day Dietary supplementation	16 weeks	▪ ↑ Japanese Orthopaedic Association symptomatic criteria ▪ Decrease in uCTX-II level and the uCTX-II/sCPII ratio	[222]
**Capsaicin cream**	Randomized, open-label series of n-of-1 trials	Radiographic knee OA *N* = 22 17 participants completed the trial.	Participants received 5% *w/w* ibuprofen gel and 0025% *w/w* capsaicin cream four times/day three treatment cycles (six paired periods) Topic administration	3 cycles, each comprising one treatment for 4 weeks, an individualized washout period (maximum 4 weeks), then the other treatment for 4 weeks	Of 22 participants, 4 (18%) had a greater response to ibuprofen, 9 (41%) to capsaicin, 4 (18%) had similar responses, and 5 (23%) were undetermined	[223]
**IA Trans-Capsaicin (CNTX)**	Randomized, Double-Blind, Placebo-Controlled Trial	Knee OA moderate-to-severe OA 45–80 years *N* = 172 *N* = 69 placebo group *N* = 33 CNTX-4975 0.5 mg group *N* = 70 CNTX-4975 1.0 mg group	Treatments: Placebo, CNTX-4975 0.5 mg, or CNTX-4975 1.0 mg Single IA injection	12 weeks 24 weeks	▪ The group treated with CNTX-4975 displayed a dose-dependent effect ▪ CNTX-4975 1.0 mg group showed a significant decrease in OA knee pain through 24 weeks ▪ The group treated with CNTX-4975 0.5 mg significantly improved pain at 12 weeks ▪ CNTX-4975 1.0 mg showed a safety profile comparable to that of the placebo throughout the study	[224]
**Pycnogenol^®^**	A randomized, double-blind, placebo-controlled trial with a parallel group design	Knee OA *N* = 37 *N* = 19 Pycnogenol (♂ + 17♀) *N* = 18 Placebo (1 ♂ + 17♀)	Patients received either a placebo or Pycnogenol pills (50 mg, three times daily) for 3 months	60 days 90 days	↓ pain, stiffness, physical function, and composite WOMAC score	[225]
**Pycnogenol^®^**	A double-blind, placebo-controlled study	Knee OA mild to moderate OA *N* = 156 I or II OA grade (X-ray analysis).	Pycnogenol group versus the placebo group 100 mg/daily Oral administration	Baseline 3 months	▪ ↓plasma free radicals and systemic inflammation ▪ ↓ C-reactive protein (CRP); ▪ ↓ Fibrinogen levels	[226]

## Data Availability

Not applicable.

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
