# Peer review of "Overview of Anti-Inflammatory and Anti-Nociceptive Effects of Polyphenols to Halt Osteoarthritis: From Preclinical Studies to New Clinical Insights"

_ijms, 2022, doi:10.3390/ijms232415861_

Round 1

Reviewer 1 Report

1. The manuscript submitted for review is well- written and very interesting from both a medical and pharmaceutical point of view. Therefore, it can trigger widespread scientific interest.

2. The authors presented a literature review from a wide time range pointing out both pros and cons of the possibility of polyphenols applications in knee osteoarthritis.

3. A few minor editorial errors in such a large article, e.g., the parenthesis ( in line 477 or the inconsistent format of some citations, e.g. 4, 6, 8, 9, 11, 12, 18 or 23, do not detract from the value of the manuscript.

Reviewer 2 Report

The authors present a study where they systematically covered and present the extensive overview of the in vitro and in vivo preclinical, but also several clinical studies focused on potential benefits of polyphenols application in OA treatment. General, this review is well organized and well structured, however, very often the construction of the sentences makes the manuscript heavy to read so I sincerely suggest working on English throughout the manuscript.  
